

**An improved method for calculating regional crop water footprint based on**
**hydrological process analysis**
**Luan Xiao-Bo[1,2,#], Yin Ya-Li[3,#], Wu Pu-Te[2,3*], Sun Shi-Kun[1,3,*], Wang Yu-Bao[1,3], Gao Xue-Rui[3],**
**Liu Jing[4]**
[1] Institute of Water Saving Agriculture in Arid regions of China, Northwest A&F University, Yangling,
Shaanxi 712100, China
[2] Institute of Soil and Water Conservation, Chinese Academy of Sciences and Ministry of Water
Resources, Yangling, Shaanxi 712100, China
[3] Key Laboratory of Agricultural Soil and Water Engineering in Arid and Semiarid Areas, Ministry of
Education, Northwest A&F University, Yangling, Shaanxi 712100, China
[4] College of Hydrology and Water Resources, Hohai University, Nanjing, Jiangsu 210098, China
# Luan Xiao-Bo and Yin Ya-Li contributed equally to this work.
* Corresponding Author:
Wu Pu-Te (gjzwpt@vip.sina.com) and Sun Shi-Kun (sksun@nwafu.edu.cn)
**Address:** Institute of Soil and Water Conservation, Chinese Academy of Sciences and Ministry of
Water Resources, Yangling 712100, Shaanxi, China





**Abstract**
Fresh water is consumed during agricultural production. With the shortage of water resources,
assessing the water use efficiency is crucial to effectively managing agricultural water resources. The
water footprint is a new index for water use evaluation, and it can reflect the quantity and types of
water usage during crop growth. This study aims to establish a method for calculating the region-scale
water footprint of crop production based on hydrological processes. This method analyzes the
water-use process during the growth of crops, which includes irrigation, precipitation, underground
water, evapotranspiration, and drainage, and it ensures a more credible evaluation of water use. As
illustrated by the case of the Hetao irrigation district (HID), China, the water footprints of wheat, corn
and sunflower were calculated using this method. The results show that canal water loss and
evapotranspiration were responsible for most of the water consumption and accounted for 47.9% and
41.8% of the total consumption, respectively. The total water footprints of wheat, sunflower and corn
were 1380-2888 $m^3/t$, 942-1774 $m^3/t$, and 2095-4855 $m^3/t$, respectively, and the blue footprint accounts
for more than 86%. The spatial distribution pattern of the green, blue and total water footprint for the
three crops demonstrated that higher values occurred in the eastern part of the HID, which had more
precipitation and was further from the irrigating gate. This study offers a vital reference for improving
the method used to calculate the crop water footprint.
**Key words**
SWAT model; Regional Scale; Water use process; Hetao irrigation district





## 1 Introduction

Human activities and climate change have serious effects on the availability of water resources (Nijssen et al., 2001; Haddeland et al., 2014). Agricultural production is major consumer of global water resources and accounts for 85% of the global blue water (surface or groundwater) consumption (Shiklomanov, 2000; Vörösmarty et al., 2010). In China, 63% of all water is used for agricultural production each year, and the area of irrigated farmland is 39.6% of the total arable land. Irrigation is the key to ensuring agricultural production (NBSC, 2016). With the rapid development of China's economy, the demand for water has increased in industrial production and in the lives of residents (Duh et al., 2008; Liu et al., 2008; Bao and Fang, 2012). Environmental pollution reduces water availability (Jiang, 2009; Schwarzenbach et al., 2010) and these changes place great pressure on regional water resources (Piao et al., 2010; Wang et al., 2014); meanwhile, climate change aggravates the situation (Elliott et al., 2014; Sun et al., 2018). With limited water resources, economic demand for water will inevitably and gradually take up the agricultural water use, which is a challenge for maintaining steady agricultural production (Chen, 2007; Khan et al., 2009), especially in the dry areas of northern China (Deng et al., 2006; Du et al., 2014). Strengthening agricultural water management and improving water use efficiency are significant aspects of handling water scarcity, and a reasonable evaluation of the water resource utilization of crop production is the premise for developing an agricultural water management plan and implementing water saving measures. Therefore, how to precisely evaluate the effective utilization ratio of current agricultural water use, improve the utilization efficiency, and reduce the negative impact of the reduction of available agricultural water is an important issue that all countries need to address Globally, this is also of vital importance for ensuring food production and reducing the pressure on water resources. The water footprint theory provides new insights and ideas to



solve these problems (Hoekstra, 2003). The water footprint is an indicator of freshwater use and can be
used to quantify water consumption throughout the production supply chain. It reflects the amount of
water and types of resources that are consumed (Hoekstra, 2011). In the agricultural sector, it can also
be used to evaluate whether a crop's water footprint is reasonable and whether it varies regionally.
Because green water can be exploited, measures need to be taken to reduce the water footprints of crop
production, especially to decrease the blue water consumption to mitigate the demand for blue water in
agriculture. The accurate and precise quantification of crop water footprints is the premise to achieving
the above goals.

Currently, many scholars have quantified various levels of crop water footprints and Hoekstra et al.

(2011) put out two main methods for calculating the crop water footprint. The first method is the crop
water requirement method. This method simulates the evapotranspiration (ET) of crops under optimal
conditions with the ET calculated by the Penman-Monteith Equation (Allen et al., 1998) and the
effective precipitation calculation provided by the United States Department of Agriculture Soil
Conservation Service (USDA SCS) (USDA, 1994). The green water ET is the smaller value of total
crop ET and effective precipitation. The blue water ET is obtained through the difference between the
total crop ET and effective precipitation. Finally, when combined with crop yields, the crop blue and
green water footprint can be calculated. The second method is the irrigation schedule method. This
method is based on an empirical formula model such as the CROPWAT model (FAO, 2010) and the
AQUACROP model (Pasquale et al., 2009). These methods can simulate crop ET throughout the
growing period according to the soil water balance under optimal or suboptimal conditions. The blue
water footprint is the smaller value of net irrigation water and the actual irrigation water requirement.
The green water ET is equal to the total ET minus blue water. Both of the above methods are based on



empirical formulas. A few scholars have attempted to calculate the region-scale water footprints, for
example, Sun et al. (2013b) used the difference between diversion and drainage to calculate the water
footprint of crop production in irrigated areas. However, these methods have certain shortages, which
are as follows:
First, the empirical methods have not determined the applicability; i.e., the method is applicable to
a field-scale or region-scale water footprint calculation. These methods calculated the field-scale water
footprint with net irrigation water considered as irrigation water, and without considering water loss
during transport or drainage, which definitely serve for crop growth. Therefore, these methods are
field-scale methods, whereas a region-scale method should include the above two losses. Presently,
irrigation water is mainly consumed by irrigated agriculture, and the current methods have not included
water loss during transport and drainage. Therefore, the field-scale water footprint calculation does not
precisely apply to irrigated agriculture, but few region-scale methods of have been established.
Second, the irrigation data in these methods are simulation values and not based on the actual
irrigation time and irrigation quota; therefore, these data cannot reflect the real situation of the local
water usage due to the incorrect simulation data. At the same time, these methods cannot distinguish
the source of the crop water, for instance, whether it is from precipitation, surface water or
groundwater.
Third, the current region-scale method has not been appropriately established. The method that
Sun et al. (2013b) used had certain limits. It included all of the water consumption, but it could not
distinguish the specific source of blue water from canal loss, field ET or groundwater. Due to its low
spatial resolution, only the water footprint of the entire irrigated area could be calculated instead of the
difference inside this area.



Agricultural production water covers the diversion - transportation - irrigation - drainage and
precise calculation of the above processes and the premise of quantifying crops' water footprints.
Currently, most studies focus on the field scale and lack systematic evaluation on the whole process of
water consumption during crop growth. To overcome this problem, this study puts forward an
improved region-scale calculation method of the crop water footprint based on hydrological process
analysis and used it to quantify the crop water footprint in HID. This method based on physical
hydrological model (SWAT), simulated the regional hydrologic cycle process, which obtained the water
consumption and the field drainage, calculated the water loss during delivery using the water
conveyance efficiency of the canal, and then quantified the region-scale crop water footprint using the
yields of the crops. This method will provide comprehensive information for the water resource
consumption process in the analysis of crop production links and improve the spatial resolution of
quantifying the crops' water footprint.
**2 Materials and methods**
**2.1 Study site**
The Hetao irrigation district (HID) is located in the middle of the Yellow River basin in western
Inner Mongolia (Fig. 1) and is one of the three largest irrigation districts in China. The HID has a
continental monsoon climate with the lowest temperature in January (average -10°C) and highest
temperature in July (average 23°C). The annual average precipitation is 180 mm and annual potential
evaporation is 220 mm. The area of the HID is $1.12\times10^4 km^2$.
Irrigation water is diverted from the Yellow River. The irrigation and drainage systems in the HID
are composed of irrigation canals and drainage ditches; the irrigation system has a general main canal
(228.9 km) and 12 main canals (total 755 km), and the drainage system has a general main ditch (227
km) and 12 main ditches (total 523 km). The main crops include wheat, corn and sunflower (Fig. 1).

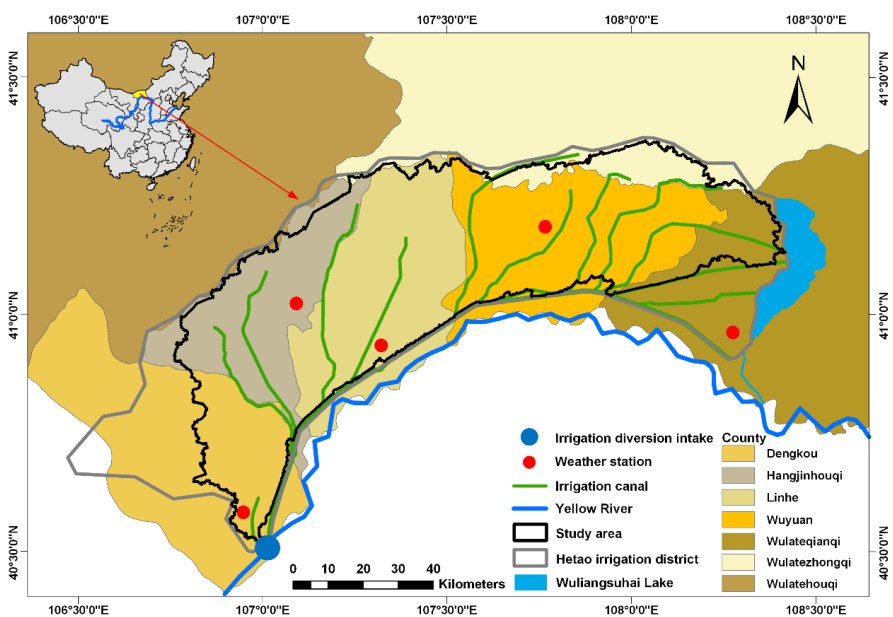

**Fig. 1.** Location of the Hetao Irrigation District (HID) in China
**2.2 Model description**
The SWAT (soil and water assessment tool) model is a semi-distributed physical hydrological
model. The model was developed by USDA Agricultural Research Center and it used climate, soil,
topography, plants and land management practices to simulate hydrologic, sediment, crop growth and
nutrient cycle. The model partitions a watershed into sub-basins by topography and then partitions the
sub-basins into hydrologic response units (HRU) based on soil type and land use to assess soil erosion,
non-point pollution, and hydrologic processes (Haverkamp et al., 2002).The water balance equation
governed by the hydrologic component of the SWAT model (Neitsch et al., 2011) is as follows:
$$SW_t = SW_0 + \sum_{i=1}^{t}\left(R_{day} - Q_{surf} - E_a - W_{seep} - Q_{gw}\right) \qquad (1)$$
where $SW_t$ is the final soil water content (mm H$_2$O), $SW_0$ is the initial soil water content (mm H$_2$O),





$t$ is the time (days), $R_{day}$ is the amount of precipitation on day $i$ (mm H$_2$O), $Q_{surf}$ is the amount of
surface runoff on day $i$ (mm H$_2$O), $E_a$ is the amount of actual ET on day $i$ (mm H$_2$O), $W_{seep}$ is the
amount of percolation and bypass flow exiting the bottom of the soil profile on day $i$ (mm H$_2$O), and
$Q_{gw}$ is the amount of return flow on day $i$ (mm H$_2$O).
**2.3 Data collection**
The data required by the SWAT model includes a digital elevation model (DEM), soil data, land
use, and hydrological and climate data (Table 1). The climate data were obtained from five weather
stations in the HID.
The water efficiency of the canal system in this model was obtained from local agricultural
administrations (AHID, 2015). To divide the sub-basins, we defined the drainage ditch as the stream
(AHID, 2015) and burn-in into the DEM, and the simulation results were verified by the discharge of
the drainage ditch.
The model generated 5 outlets and 73 sub-basins, and the measured data of the first outlet in the
study area was obtained. Therefore, this study chose the area controlled by this outlet as the study area.
The crops' yields (wheat, corn and sunflower) required for the calculation of the water footprint was
obtained from the Statistical Yearbook of the local agricultural administrations (AHID, 2015).
Table 1 Data used in the study and the resources.

| Dataset | Data description | Resolution | Data sources |
|---|---|---|---|
| DEM | — | 30×30 m | Geospatial Data Cloud (CAS, 2009a) |
| Soil | Soil type map, Soil physical and chemical properties | 1:1000000 | China Soil Scientific Database (CAS, 2009b) |
| Land use | — | 1:100000 (2010) | Data Center for Resources and Environmental Sciences (CAS, 2010) |





| Weather | Precipitation, Wind speed, | Daily | China Meteorological Data Network |
|---|---|---|---|
| | Solar radiation, | (1980-2012) | (NMIC, 2015) |
| | Maximum temperature, | | The Administration of Hetao Irrigation |
| | Minimum temperature, | | District (AHID, 2015) |
| | Relative humidity | | |
| Hydrologic | Stream map, | Monthly | The Administration of Hetao Irrigation |
| | Discharge | (2003-2012) | District (AHID, 2015) |
| Crop | Dates of plant and harvest, | — | The Administration of Hetao Irrigation |
| parameter | Dates of irrigation, | | District (AHID, 2015) |
| data | Irrigation quota | | |


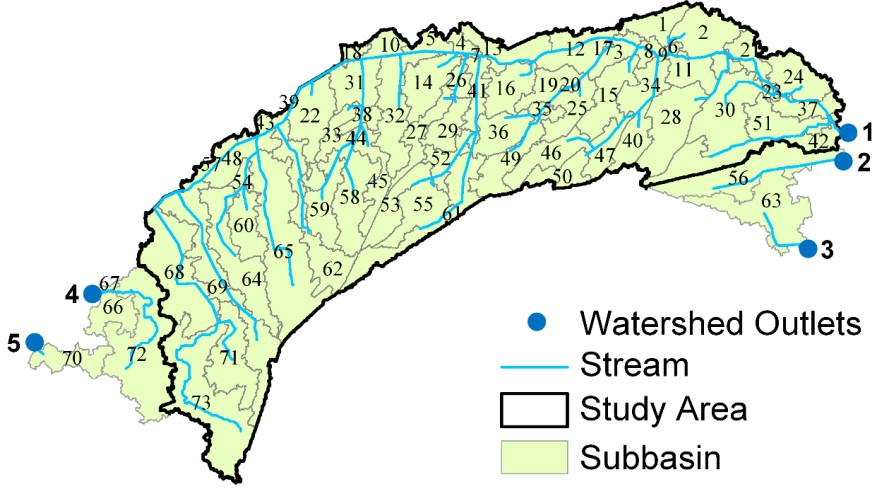


**Fig. 2.** Subbasins and study areas
**2.4 Calibration and validation**
The Sequential Uncertainty Fitting (SUFI-2) algorithm in SWAT-CUP was applied for calibration
and validation (Abbaspour et al., 2007; Abbaspour, 2012) by comparing the simulated stream discharge
from the model with the measured discharge data. The global sensitivity analysis integrated within
SUFI-2 was used to evaluate the hydrologic parameters for the discharge simulation and then the





optimal simulation is established by adjusting the sensitivity parameters and through multiple iterations.
The calibration period was from 2006-2009, and the validation period was from 2010-2012.

For calibration and validation analyses, the monthly measured discharges were compared with the

simulated discharge data and the model performance was evaluated using the coefficient of
determination ($R^2$), Nash efficiency coefficient (NSE) (Nash and Sutcliffe, 1970; Moriasi et al., 2007)
and percent deviation (PBIAS) (Gupta et al., 1999). The calculation formula is as follows:
$$R^2 = \frac{\left[\sum_{i=1}^{n}\left(Q_m - \overline{Q}_m\right)\left(Q_s - \overline{Q}_s\right)\right]^2}{\sum_{i=1}^{n}(Q_m - \overline{Q}_m)^2 \sum_{i=1}^{n}(Q_s - \overline{Q}_s)^2} \qquad (2)$$

$$NSE = 1 - \frac{\sum_{i=1}^{n}(Q_m - Q_s)^2}{\sum_{i=1}^{n}(Q_m - \overline{Q}_m)^2} \qquad (3)$$

$$PBIAS = \frac{\sum_{i=1}^{n}(Q_m - Q_s)}{\sum_{i=1}^{n}Q_{m,i}} \times 100 \qquad (4)$$

where $Q_m$ is the measured data, $\overline{Q}_m$ is the mean of the measured data, $Q_s$ is the model
simulation data, and $\overline{Q}_s$ is the mean of the model simulation data.

$R^2$ measures the simulated and measured values of goodness. The closer the value is to 1, the

higher the agreement is between the simulated and measured discharge. The NSE is widely applied in
hydrologic models that range from negative infinity to 1 with 1 being the ideal value. The PBIAS
assesses the average deviation of the simulated values from observed values with 0 as the ideal value,
and a positive (negative) PBIAS value shows an underestimation (overestimation) bias of the simulated
variable compared to the measured variable. The monthly model data simulation results can be
classified as satisfactory if $R^2 > 0.6$, NSE > 0.5 and PBIAS < ±25 and can then be used for further



analysis (Moriasi et al., 2007).
The SWAT-CUP parameter sensitivity analysis procedure showed that the CN2, ESCO,
GW_REVAP and ALPHA_BF parameters were more sensitive. In this study, the R2, NSE, and BIAS
for the measured and calibration period were 0.77, 0.65 and 17, respectively; and the R2, NSE, and
PBIAS for the validation period were 0.68, 0.61 and 21, respectively(Luan et al., 2018). The model
simulation result can be classified as satisfactory (Moriasi et al., 2007). Therefore, the results
demonstrated that the SWAT model was applicable in HID for future hydrologic process assessments.
**2.5 The region-scale water footprint calculation method**
Based on the water footprint theory framework provide by Hoekstra et al. (2011), this study
suggests a new way of quantifying the region-scale water footprint of crop production (Fig. 3).

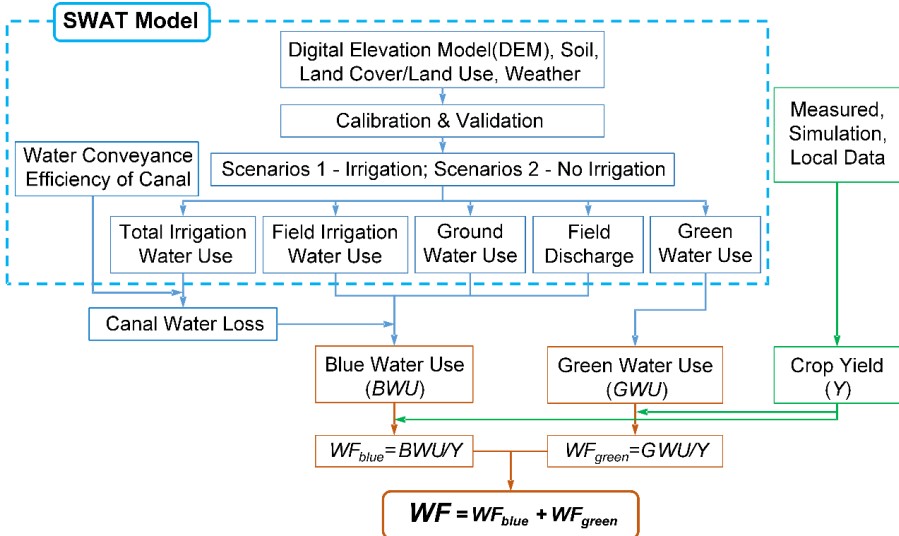


**Fig. 3.** The flowchart for calculating the region-scale water footprint
In this study, green water consumption is the ET produced by the consumption of precipitation
during crop growth. Blue water consumption includes canal water loss during delivery, the ET
produced by consumption of irrigation water and groundwater for crops growth, and the drainage in the





fields. To calculate the canal water loss, an extra model needs to be established according to the HID
situation, and the other can be simulated and obtained by the SWAT model.
**2.5.1 Calculation of water consumption factors in the fields**
Water consumption in the fields consists of 4 parts including the ET of precipitation, irrigation
water, groundwater utilized by crops, and field drainage. This study set up two scenarios and calculated
the above water consumption by changing the sources of water in the SWAT model. In scenario 1 (S1),
water consumption was derived from precipitation and irrigation water in the fields (irrigation systems
and irrigation quotas are based on local irrigation methods), i.e., the actual situation of crop water use.
In scenario 2 (S2), water consumption was only derived from precipitation without irrigation. In S1,
after calibration and validation of the model, and by modifying the crop water management data,
removing irrigation water, and simulating again, the results in S2 could be obtained. Then, the results
were calculated using two simulations, specifically, modifying the single variable to observe the
corresponding result. The calculation formula is as follows.
$$WF = WF_g + WF_b = \frac{W_g}{Y} + \frac{W_b}{Y} \tag{5}$$
$$W_g = ET_{s2} - Q_g \tag{6}$$
$$W_b = Q_c + Q_f + Q_g + Q_d \tag{7}$$
$$Q_c = I_t - I_f \tag{8}$$
$$Q_f = ET_{s1} - W_g \tag{9}$$
where $WF$ is the water footprint of crop production (m³/t), $WF_g$ is the green footprint (m³/t), $WF_b$
is the blue water footprint (m³/t), $W_g$ is the green water consumption during the crop growth period
(m³), $W_b$ is the blue water consumption during the crop growth period (m³), $Y$ is the crop yield (t), $ET_{s1}$
is the crop actual ET during the crop growth period in Scenario 1 (m³), $ET_{s2}$ is the crop actual ET





during the crop growth period in Scenario 2 (m³), $Q_g$ is the amount of groundwater that rises to the soil
plow layer (m³), $Q_c$ is the amount of water loss in the canal system (m³), $Q_f$ is the ET of field irrigation
water (m³), $Q_d$ is the field discharge (m³), $I_t$ is the amount of total irrigation water diversion (m³), and $I_f$
is the actual amount of water irrigated in the field (m³).

### 2.5.2 Calculation of water loss during delivery

Water loss during transportation occurs in the canal and is an important part of blue water
consumption of the crops growth. Because of the complexity of the irrigation canal system and the lack
of hydrological data (lack of water conveyance efficiency of the branch canal and lower canal), we
generalized the irrigation area into a similar rectangle model (Fig. 4). Each rectangle is the area
controlled by each main canal, which is represented by the central line. The natural canal system is
divided into two parts when calculating the water loss of the canal system. Part A is the loss of the
general main canal and the main canals, and the part B is the loss of the rest of the canal system
including the branch canals, lateral canals, field canals, and sub-lateral canals.
The water loss in part A could be calculated as follows: divide the main canal by equidistance (10
km) and then calculate the water loss of each section, which was produced by local and downstream
water of which local water accounted for a small amount and the rest belonged to the downstream.
Therefore, the local accurate water loss should include this section and upstream sections. We assumed
local water loss to the midpoint of each canal. In ArcGIS, we used a Kriging interpolation to obtain the
water loss figure of part A.
Water loss in part B could be calculated as follows: the water loss of the other canals below the
main canal divided by the area controlled by each main canal and the water loss per unit area controlled
by the corresponding canal could be obtained. Then, the water loss per unit area controlled by each


main canal could be obtained. The data of parts A and B are calculated using the Space analysis tool in
ArcGIS 10.1 software to obtain the distribution map of the water loss in the drainage system.
The formulas are as follows:
$$W_A = I_t \times \left(1 - k_g \times k_m\right) \tag{10}$$
$$S_{ji} = \frac{S_j}{i} \tag{11}$$
$$W = \frac{W_A \times k_j}{i \times S_{ji}} \tag{12}$$
$$Q_n = W \times \left(\frac{1}{i} + \frac{1}{i-1} + \frac{1}{i-2} + \cdots + \frac{1}{i-(n-1)}\right) \quad n \in \left(1,2,3,\cdots,i\right) \tag{13}$$
$$W_B = Q_c - W_A \tag{14}$$
$$Q_j = \frac{W_B \times k_j}{S_j} \tag{15}$$
where $W_A$ is the amount of water loss in part A (m³), $I_t$ is the amount of total irrigation water diversion
(m³), $k_g$ is the water conveyance efficiency of the general main canal, $k_m$ is the water conveyance
efficiency of the main canal, $S_j$ is the area controlled by the $j$th main canal (ha), $i$ is the number of the
equidistance section of the $j$th main canal, $S_{ji}$ is the area per section controlled by the $j$th main canal
(ha), $k_j$ is the ratio of the diversion volume of the $j$th main canal to the total diversion, $W$ is the water
loss per unit area of the section of the $j$th main canal in part A (m³/ha), $Q_n$ is actual the amount of water
loss per unit area of the section of the $j$th main canal (m³/ha), $W_B$ is the amount of water loss in part B
(m³), and $Q_j$ is the water loss per unit area of the $j$th main canal (m³/ha).




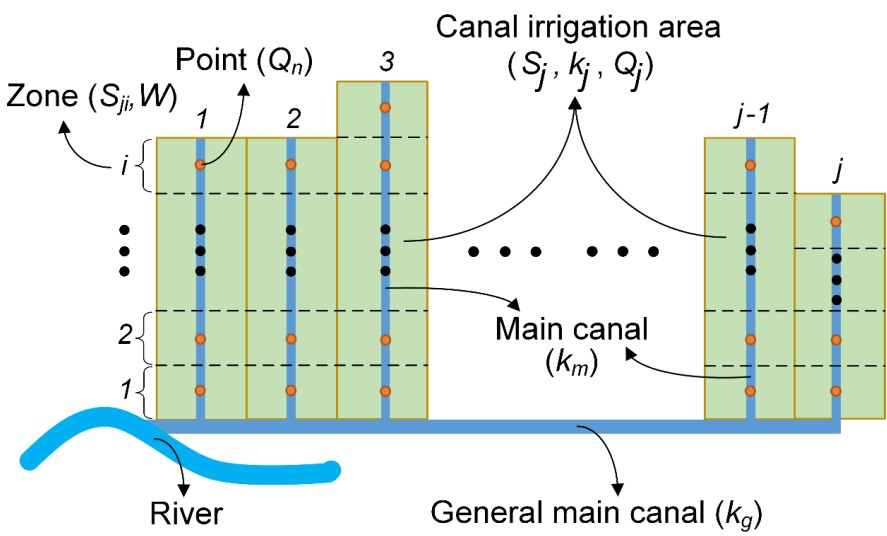


**Fig. 4.** Generalized model of the irrigation area
**3 Results**
**3.1 Analysis of the process of crop production and the quantification of hydrological**
**elements in each link**
Fig. 5 shows the average water input and consumption of the study area in the process of water
diversion, transportation, irrigation and drainage from 2006 to 2012. In HID, the water input for
irrigation for the three crops in the study area was 3177 $Mm^3$, water loss during transportation in the
canals was 1652 $Mm^3$, the actual field irrigation water was 1525 $Mm^3$, precipitation in the farmland
was 510 $Mm^3$, the actual ET of the farmland was 1442 $Mm^3$, the discharge was 352 $Mm^3$, and the
groundwater was not considered because the consumption was less than 5%. When inputting water into
the farmland, irrigation and precipitation accounted for 74.9% and 25.1%, respectively; however, when
consuming water, the discharge took up 47.9%, 41.8% and 10.3%, respectively. Irrigation was the main
water source in the irrigated district, and the water loss in the canals and actual ET were the main water
output in the irrigated district.



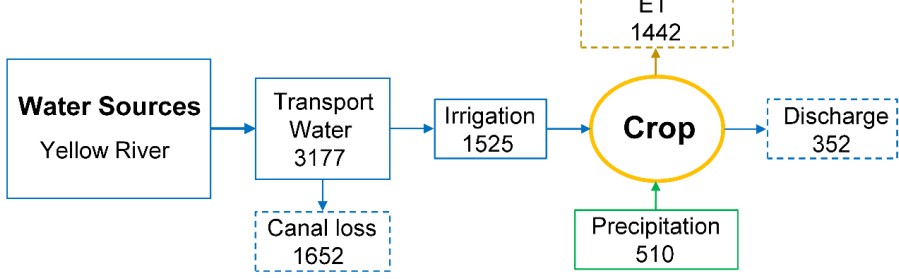

**Fig. 5.** The amount of water during crop growth (Mm³)

Green water is the precipitation used for crop growth; therefore, the green water footprint is highly

correlated with precipitation in its growth period. Wheat's growth period is from April to July, whereas

that of corn and sunflower is from May to September. During the growth period of wheat, the mean

precipitation from 2006 to 2012 was 108.9 mm, and for corn and sunflower, the corresponding mean

precipitation was 176.1 mm. The green footprint of wheat during the growth period was lower than that

of corn and sunflower because of the lower mean precipitation in the wheat growth period. The green

water consumption of corn was close to the value of sunflower. The green water consumption of wheat,

corn and sunflower were 895 $m^3$ $ha^{-1}$, 1441 $m^3$ $ha^{-1}$ and 1419 $m^3$ $ha^{-1}$ (Fig. 6 a1, b1, c1), respectively.

Meanwhile, green water consumption in the high precipitation area was larger, for instance, the

precipitation during the wheat growth period in Wuyuan reached 116.3 mm, and the green water

consumption in this region was the largest (up to 995 $m^3$ $ha^{-1}$). In the growth period of corn and

sunflower, the precipitation in Wulateqianqi reached 199.4 mm, and the green water consumption in

this area was again the largest, reaching 1785 $m^3$ $ha^{-1}$ and 1765 $m^3$ $ha^{-1}$, respectively.

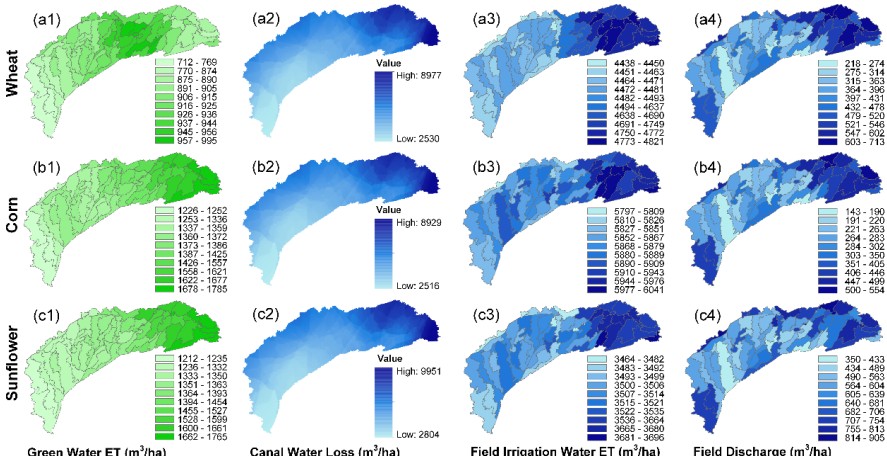


**Fig. 6.** Spatial distribution of the different water consumption of three crops (m³ ha⁻¹)


Blue water is surface water and groundwater used for crop growth. In blue water consumption, the
farther away from the watershed inlets the longer the canal was and the larger the water loss of the
three crops. Northeast of the irrigation area (parts of Wuyuan and Wulateqianqi) and due to the far
distance from watershed inlets, canal water loss in these places was much higher than that in other
areas, and the maximum canal water loss of wheat, corn and sunflower reached 8977 m³ ha⁻¹, 8929 m³
ha⁻¹ and 9951 m³ ha⁻¹, respectively. The different amount of canal water loss was caused by the
difference of water loss in the unit area, at 4778 m³ ha⁻¹, 4753 m³ ha⁻¹ and 5297 m³ ha⁻¹, respectively.
The actual ET and the discharge of the three crops was higher in the east than in the west, which
was due to the higher evaporation in the east than in the west. Meanwhile, Fig. 6 shows that the actual
ET in the field was complementary with discharge. The higher the actual ET, the smaller the discharge
and vice versa.
**3.2 The regional green water footprint of crop production**
The green water footprint of the crops is produced by precipitation during crop growth. The spatial
difference of the green water footprints of wheat, corn and sunflower in HID was obvious (Fig. 7). It




can be seen from the figure that the overall distribution of the green water footprint of the three crops
was higher in the east than it was in the west. However, the distribution of green water footprints was
somewhat different for each crop. Wheat had the largest green water footprint in Wuyuan (197 $m^3$ $t^{-1}$)
and the lowest in Dengkou (132 $m^3$ $t^{-1}$). Corn had the largest green water footprint in Wulateqianqi (186
$m^3$ $t^{-1}$) and the lowest in Hangjinhouqi (119 $m^3$ $t^{-1}$), but in Dengkou, it was approximate to that in Linhe,
ranging from 133 to 139 $m^3$/t. Sunflower had the largest green water footprint in Wulateqianqi (538 $m^3$
$t^{-1}$) and the lowest in Linhe (325 $m^3$ $t^{-1}$). The green water footprint of crop production also varied across
crops. The largest average green water footprint in HID was sunflower, followed by wheat and corn.

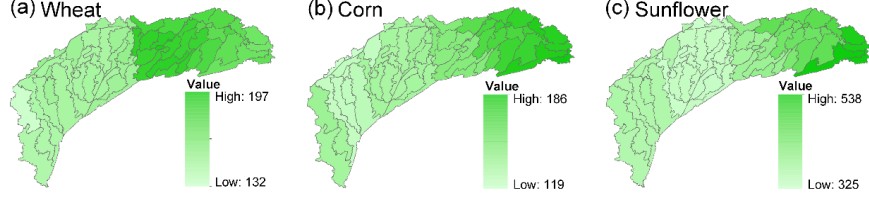


**Fig. 7.** The spatial distribution of the green water footprint of crop production in the HID ($m^3$ $t^{-1}$)
**3.3 The regional blue water footprint of crop production**
The blue water footprint of the crops is produced by blue water that is consumed during crop
growth. The blue water consumption during crop growth mainly includes the loss during transportation,
ET and field drainage. Fig. 8 shows the spatial variability of wheat, corn, and sunflower in HID. The
overall distribution of the total water footprint of the three crops was higher in the east than in the west
and higher in the north than in the south. However, the specific distribution was somewhat different for
each crop. Wheat had the largest blue water footprint in Wulateqianqi (2714 $m^3$ $t^{-1}$) and the lowest in
southern Linhe (1233 $m^3$ $t^{-1}$). Corn had the largest blue water footprint in northern Wulateqianqi (1588
$m^3$ $t^{-1}$) and the lowest in southern Hangjinhouqi (820 $m^3$ $t^{-1}$). Sunflower had the largest blue water
footprint in northern Wulateqianqi (4317 $m^3$ $t^{-1}$) and the lowest in southern Linhe (4317 $m^3$ $t^{-1}$). The
blue water footprint of crop production also varied across crops. The largest of the average blue water
footprint in the HID was sunflower, followed by wheat and corn.

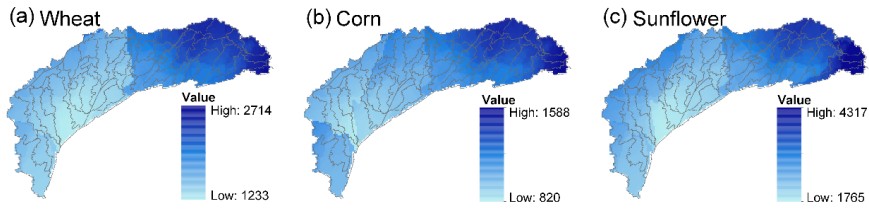

(a) Wheat    (b) Corn    (c) Sunflower

Value High: 2714  Low: 1233

Value High: 1588  Low: 820

Value High: 4317  Low: 1765

**Fig. 8.** The spatial distribution of the blue water footprint of crop production in the HID ($m^3$ $t^{-1}$)
**3.4 The regional total water footprint of crop production**
The total water footprint of crop production consists of both blue and green water footprints
during the crop growth period. Fig. 8 shows the total water footprint of crop production and spatial
variability of wheat, corn, and sunflower in HID. The overall distribution of the total water footprint of
the three crops was higher in the east (Wulateqianqi and Wuyuan) than it was in the west (Dengkou),
followed by the central region (Hangjinhouqi and Linhe) and was higher in the north than in the south.
However, the specific distribution was somewhat different for each crop. Wheat had the largest total
water footprint in the east (Wulateqianqi, 2888 $m^3$ $t^{-1}$) and the lowest in southern Linhe (1380 $m^3$ $t^{-1}$).
Corn had the largest total water footprint in the east (Wulateqianqi, 1774 $m^3$ $t^{-1}$) and the lowest in
southern Hangjinhouqi (942 $m^3$ $t^{-1}$). Sunflower had the largest total water footprint in the east
(Wulateqianqi, 4885 $m^3$ $t^{-1}$) and the lowest value was in southern Linhe (2095 $m^3$ $t^{-1}$). The total water
footprint of crop production also varied across crops. The largest of the average total water footprint in
the HID was sunflower, followed by wheat and corn. The blue water footprint of wheat, corn and
sunflower accounted for 89%, 87% and 86% of the total water footprint, respectively.

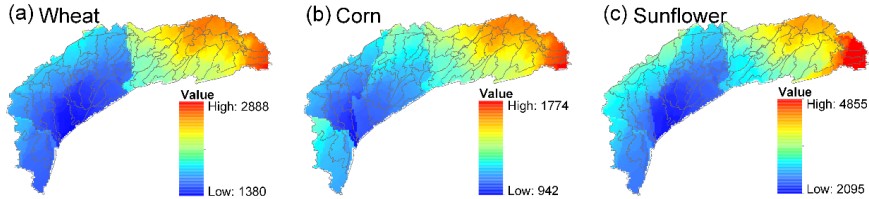


**Fig. 9.** The spatial distribution of the total water footprint of crop production in the HID (m³/t)


**4 Discussion**
**4.1 Methods of calculating crop production water footprints——region scale and field scale**

In this paper, the calculation method of calculating crop production water footprints is divided into

the field and region scales, according to the calculation boundary of water consumption in crop growth.

The field-scale water footprint is composed of the transpiration of crops and the evaporation of

soil, and the water loss during transportation is not included. Current studies had different geographical
scales, such as a global scale (Mekonnen and Hoekstra, 2011), a national scale, such as Europe
(Vanham and Bidoglio, 2013) and China (Zhao, 2009), and a regional scale, such as Beijing (Sun,
2013a), Cremona province (Bocchiola, 2015) and Hetao (Luan et al., 2018); however, they all
calculated the crop production water footprint of the field scale. These studies were based on empirical
formulas, which could be divided into two methods. The first method is the irrigation schedule method,
such as CROPWAT (Mekonnen and Hoekstra, 2011), CropSyst (Bocchiola et al., 2013), the EPIC
model (Williams et al., 1989; Shi et al., 2017), the GEPIC model (Liu et al., 2007), and the
AQUACROP model (Chukalla, 2015; Zhuo 2016). The other method is the crop water requirement
method (Cao et al., 2014; Sun et al., 2013c). The calculation of crop ET in these methods was based on
the full satisfaction of the crop water requirement, there is no water deficit, and the actual soil water
content was not taken into consideration. Therefore, the results did not reflect the actual water
consumption of the crops, and the water footprint of the crop production in the field scale cannot





distinguish the source of blue water consumption from the surface water or groundwater.
The region-scale water footprint calculation method considered all of the water consumption
related to crop growth from the water source to the field. It not only included the ET from the field but
also the water loss during transportation in the canal system and the water loss discharged out of the
region. The blue water was consumed for crop growth and thus had to be included in the calculation of
the water footprint. This was also the definition of crop water consumption in the crop production
water footprint concept, which included all of the processes related to crop production, such as storage
and transportation (the water that ran to other basins or seas such as the discharge out of the region
instead of running back to the former basin) (Hoekstra, et al., 2011). The water footprints of the whole
area irrigated by the canal system could be calculated by the region-scale method. To date, few studies
have examined a region-scale water footprint. Sun et al. (2013b) calculated the regional water footprint
in HID; however, the calculation was merely based on the principle of water balance and calculated the
blue water consumption of the whole region based on the difference of water diversion and discharge in
the region without distinguishing the specific parts of blue water loss.

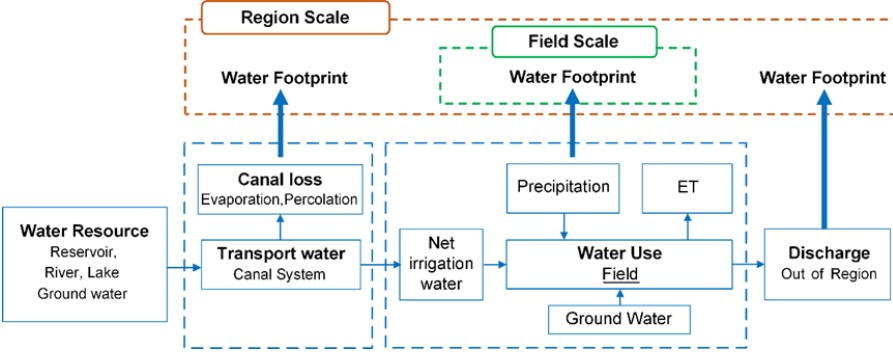


**Fig. 10.** The different scales of calculating water footprints
**4.2 Comparison of the applicability of two methods**





The applicable conditions of the two methods of calculating water footprints are different. In
terms of the calculation boundary, the calculation of the green water footprint is the same, whereas the
calculations of the blue water footprint are different. Fig. 11 illustrates the water sources and use
conditions of two types of agriculture. The rainfed agriculture depends on precipitation (green water)
and groundwater (blue water), and the water consumption mainly includes ET. While irrigation
agriculture relies on surface water, groundwater and precipitation, water consumption includes ET,
transport loss and discharge. Therefore, the field-scale method is suitable for calculating the water
footprint of rainfed agriculture, whereas the region-scale method applies to the calculation of the
irrigation agriculture water footprint.
Currently, irrigated farmland occupies 39.6% of the total arable land in China (NBSC, 2016).
Globally, irrigated area accounts for 20.6% of all arable land (FAO, 2016). Overall, the yields of
irrigation agriculture are much higher than that of rainfed agriculture. If the water footprints of
irrigation agriculture are calculated by the field-scale method without considering water loss during
transportation or discharge, the calculated values are smaller than the actual values, and the actual
water footprints of irrigation agriculture cannot be precisely assessed. This is also the deficiency of the
current crop production water footprint studies because most studies have adopted the field-scale
method. Therefore, using the region-scale method to calculate the crop water footprint, especially in
irrigation agriculture, is the basis for a comprehensive and accurate evaluation of a crop production
water footprint in China and other regions or countries.



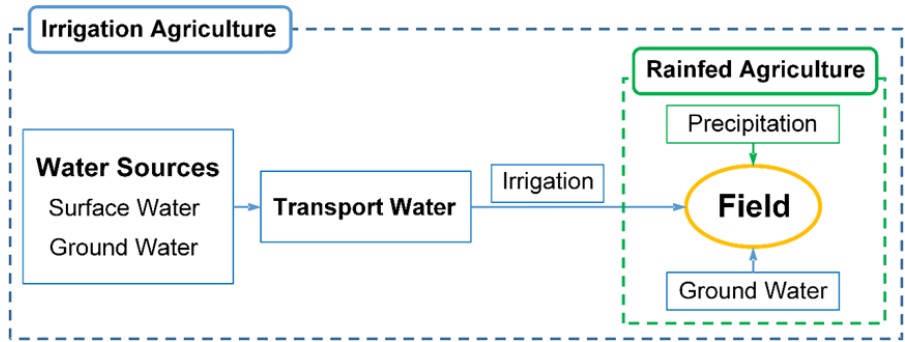


**Fig. 11.** Irrigation agriculture and rainfed agriculture


**4.3 The methods of calculating region-scale crop production water footprints**


In this study, we proposed an improved calculation method of the region-scale crop production
water footprint. The method based on the hydrological model (SWAT model), which used the irrigation
canal water use coefficient, calculated all of the water consumption in the process of crop growth by
area (Hetao irrigation area) such as green water consumption, blue water in conveying process
consumption, and irrigation and drainage in the field. The SWAT model could be used to simulate the
regional hydrologic cycle and its simulation results could calculate the water use process during the
crop growth period such as irrigation, precipitation, groundwater, ET and drainage. Then, combined
with the water conveyance efficiency of the canal, the water canal loss during transportation could be
calculated. In addition, this method could calculate the use of groundwater during the crop growth
period, and therefore, blue water could be divided into surface water and groundwater for an additional
accurate analysis of water sources for crop growth. To date, many scholars have conducted a few
corresponding studies (Mekonnen and Hoekstra, 2010). Therefore, this method can calculate water use
during the crop growth period and then more precisely calculate the blue, green and total crop water
footprints.
In HID, the canal water loss accounted for 47.9% of all water consumption, which is one of the



main water consumption components during the crop growth period. Therefore, it is necessary to
calculate the crop water footprints in irrigated areas using the regional scale. The water footprints of
three crops (wheat, corn and sunflower) in HID and calculated by this method are 1380-2888 $m^3$ $t^{-1}$,
942-1774 $m^3$ $t^{-1}$, and 2095-4855 $m^3$ $t^{-1}$, respectively. These values are higher than the results calculated
by the field-scale method. Cao et al. (2014) calculated the mean crop water footprints of China
irrigation agriculture from 1998 to 2010 in which the mean total water footprint of many crops in the
Inner Mongolia autonomous region (including HID) was 1556 $m^3$ $t^{-1}$. Sun et al. (2013b) used the
region-scale method and the water balance principle to calculate the average water footprint of HID
and it was 3.91 $m^3$ $kg^{-1}$ in which blue water accounted for 90.9% and green water accounted for 9.1%.
This result was the average water footprint of many crops, and the value was approximate to our results
for the blue water of wheat, corn and sunflower and accounted for 89%, 87% and 86%, respectively.
However, Sun et al. (2013b) could not distinguish each crop or illustrate the difference of spatial
distribution.

The region-scale method proposed in this paper not only applies to water footprints of irrigation

agriculture but also applies to the calculation of rainfed agriculture. If there is only natural precipitation
without irrigation in the study area, irrigation can be excluded in the SWAT model to simulate the water
circle in the field with rainfed conditions to calculate the field-scale water footprints of crop production.
Therefore, this study method is suitable for two scales.

There are limitations to this approach. The method needs more data types (for instance, DEM,

land use, soil and climate data, hydrological data, and crop management), and high-precision data is
required, which are difficult to obtain. This method does not apply to areas without the above data.
**5 Conclusions**





In this study, we proposed an improved region-scale method for calculating crop water footprints.
This method is based on the hydrological model (SWAT model), combined the irrigation parameters of
the irrigation area (water conveyance efficiency of canal), and calculated the crop production water
footprints.
The method can analyze the process of water use during the crop growth period, including
irrigation precipitation, groundwater, ET and drainage, for a more comprehensive calculation of water
consumption during the crop growth period and more precisely quantify crop production water
footprints. The method can be applied to calculate the crop production water footprint at both the field
and region scale. In HID, the main water consumption occurs during the crop growth period; the canal
water loss was 1652 Mm$^3$ and ET in the field was 1442 Mm$^3$, which accounted for 47.9% and 41.8% of
the total consumption, respectively.
Based on this method, the total water footprints of three crops (wheat, corn and sunflower) in HID
were 1380-2888 m$^3$ t$^{-1}$, 942-1774 m$^3$ t$^{-1}$, and 2095-4855 m$^3$ t$^{-1}$. In terms of spatial distribution, the
values were higher in the east than they were in the west. The spatial distributions of blue and green
water footprints were similar, and the blue water footprint accounted for more than 86% of the total
water footprint.
Green water consumption was directly related to precipitation in the crop growth period. Less
precipitation in the growth period of wheat led to less green water consumption and blue water
consumption accounted for 93.1%. For corn and sunflower, blue water consumption accounted for 89.7%
and 90.1%, respectively. For blue water consumption, water loss during transportation increased with
the increasing distance of the canals, and the farther away from the watershed inlets they were, the
more water was lost.



**Acknowledgements:**
This work is jointly supported by the National Natural Science Foundation of China (51409218;
51609063), the National Key Research and Development Program of China (2016YFC0400201), and
Science and Technology Integrated Innovation Project, Shaanxi Province of China
(2016KTZDNY-01-01),the Open Research Fund of the State Key Laboratory of Simulation and
Regulation of Water Cycle in River Basin at the China Institute of Water Resources and Hydropower
Research (IWHR-SKL-201601) and Young Scholar Project of Cyrus Tang Foundation.
**Author Contributions:** Pute Wu, Shikun Sun and Yubao Wang designed the study. Xiaobo Luan, Yali
Yin and Jing Liu did the literature search and data collection. Shikun Sun, Xiaobo Luan and Xuerui
Gao managed and analyzed the data. Suikun Sun and Xiaobo Luan drew the figures and wrote the
paper. All authors discussed and commented on the manuscript.



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
