# Peer review of "An improved method for calculating regional crop water footprint based on hydrological process analysis"

_Hydrology and Earth System Sciences, 2018_

## Referee Comment (RC1) · Anonymous Referee #1 · 25 Apr 2018

In this manuscript the authors enhance the Water Footprint method to a regional scale. For that purpose, the water losses in the irrigation water distribution system are included into the WF (blue water). A procedure for the quantification of water losses in the canal system is introduced. The topic addressed in this manuscript is relevant and has certain novelty for the water footprint assessment. Nevertheless, several minor and major aspects need to be improved. The overall calculation approach needs to be checked, since some calculation steps are contradictory to the common water footprint method. The grammar is very poor throughout the whole paper and has to be improved.

General comments: Water loss from the canal system and application on the fields is calculated as water consumption. This is not correct, because leached water recharges the local groundwater aquifers and, thus, contributes to the water availability. In that term, it is not consumed. Therefore, water consumption equals the actual evapotranspiration (for the agricultural production), while water losses (leaching from the fields) is subtracted from the water used for the irrigation. Therefore, I do not understand why you calculate it as water footprint. I can understand if you state that water is recharged in another aquifer, however the groundwater aquifers are connected. Please consider that issue with the water losses and the spatial aspect of water losses. The second scenario is not plausible, because, as stated later on in this paper, 75% of the water input comes from the irrigation, that is why this is not a red-fed agriculture and stopping all irrigation will lead to a total yield loss. Therefore, this scenario cannot exist in reality. If you want to develop a scenario, please consider, for example, deficit irrigation. Nevertheless, I do not see the necessity of the second scenario for your paper. The methodology for calculating WF on the field level (so, for the rain-fed agriculture) already exists and is broadly applied. Therefore, there is no novelty in your approach concerning that aspect. Furthermore, the results of the second scenario are missing in the section Results. Regarding the term ET, it would be good to distinguish between the potential and actual ET. I would appreciate if you specify it in the text. I would appreciate if you also address the importance of efficiency of the water distribution systems in the Discussion Please improve the grammar throughout the whole paper. Your sentences are sometimes built in a strange way and are difficult to understand. I recommend to ask a native speaker to review the paper.

Specific comments and technical corrections: L21: effectively managing -> effectively manage. L22: It is not correct to state that WF is a new method; it is used in the scientific community since decades. I would recommend deleting this statement. Abstract: please address that you are quantifying the WF in terms of blue and green water, because there are other methods, for example impact assessment using the AWARE, WSI and WAVE models, but also the grey water footprint. L26: do you mean with the

term "underground water"? Groundwater use? Please use another term. L34: further -> further away. L44: to ensuring -> to ensure. L54: utilization of crop production -> for crop production. L57: What do you mean with "reduce the negative impact of the reduction of available agricultural water"? Please check and correct this sentence. L58: Globally -> globally. L62: Please address and explain here the terms green, blue and grey water instead of the term "types of resources". L72: This data is not necessarily provided by USDA, it is available in other sources. Of course, you used this data source for your case study, but here it is better to delete the reference to the USDA. L76: Please indicate, that the WF is calculated per kg or ton of crop. L68-81 – please insert the equations for the calculation of green and blue WF. L80 – Please explain the terms net irrigation water and the actual irrigation water requirement. L89: I agree that the existing methods do not consider the water losses in canals, but the water leached from the fields (if I correctly understand the term "drainage water") is actually considered. Therefore, please revise this statement. L95: The term "irrigation quota" is not clear. Do you mean the irrigation demand? Please add the definition and probably also use another term throughout the whole paper. L96: I would not define it as incorrect, rather not complete. L97: that is not true, precipitation and irrigation water are distinguished, that is why it is possible to distinguish between the green and blue water. The irrigation water is calculated as the amount of water to meet the part of the crop water requirement, which cannot be fulfilled by the precipitation. Concerning the groundwater, you are correct: surface water and groundwater are evaluated together for the blue water calculation. L99-101: iI is not clear how it is different from the other limitations you mentioned above. L106-107 – this a repetition. L109 – 115 – the sentences are too long and difficult to understand. Please rewrite this section. L121-122: Since it has monsoon climate, the annual average values for precipitation and ET do not provide any valuable data. Consider providing average values for the dry season and monsoon season. Fig 1: Please use another word for the areas (Dengkou etc), e.g. districts in the legend, because the term "country" is misleading. L125: Could you please explain more the differences between canals and ditches? L170-172: I do not

see a benefit of writing the formulas 2, 3 and 4 into the text. I recommend deleting them. L175-176: The explanation of the $R^2$ is not needed, since it is a very common parameter in statistics. L166-189: This section includes information on the uncertainty analysis, which is a bit confusing. It distracts the reader from the actual content of the paper. It will be better to put this data into the supplementary material and only mention that the results are satisfactory . L203: What exactly do you mean with the term "irrigation water in the fields"? Surface water from the irrigation channels? Please use another term, otherwise it is not clear what you mean. L205: I do not understand how this scenario should work in reality. If the fields are irrigated, this means, that there is not enough precipitation to meet the crop water requirement. That would mean that in your scenario 2 the crops do not have enough water. So, do you also consider the yield loss? Because you cannot get the same yield under such different conditions. Equations 5-9 – please check the formatting. Eq.5 – this equation is obvious, you can describe it in the text (total WF is the sum of the green and blue WF), but you don't need an equation for that. Eq. 6- please provide more explanation. It is not clear why the groundwater, which raises to the soil plow layer, is included into the green WF calculation. Eq.7: $Q_d$ – same as for the water loss in the canal – this is not a loss because the water recharges the GW aquifer, so actually the equation should be $Q_f$-$Q_d$. Again, including $Q_g$ into the water consumptino is not clear. Fig4 – please include explanation of Acronyms and indices into the name of the table. Could you also show the part A and part B on the figure? Eq. 10-15 – check formatting of the numeration of the equations. It is difficult to follow the equations. It would help to understand, if you split them into calculation for the part A and part B and include the explanations of the indices directly after each equation. More description of the whole calculation path is needed to follow the calculation procedure. L253: Do you mean "sections"? L255: $Q_n$ is the actual amount. . .. L270: I do not understand for which parameters these rates are. Precipitation, irrigation on field and canal loss? L290: Since you didn't consider the groundwater irrigation, please indicate here, that the blue WF includes only the surface water irrigation. L299-300: I do not see this correlation in your results.

Actually, there is rather more field discharge by larger ET, if I understand your figures correctly. Please check this data. It is also interesting whether there are same irrigation techniques applied on the whole study area. Because water loss on the field depends on the irrigation method applied. Please address this aspect. L302: This sentence is a repetition, please delete. L322-325 and L310-311: These statements are obvious, because these three crops have different crop water requirement. Please consider deleting these sentences. L345: Do you mean "water footprints on the regional scale and field scale"? L346: Method for. L350-353: I understand what you want to say, but this sentence is misleading, because you firstly states, that the studies are on national, regional etc scale, but then says that the studies are on the field scale. Please change the rephrase to make it more understandable. L350-362: There is too much text explaining the methods. I recommend deleting L353-360 or insert them into the introduction and refer to it here. L366-370 –This text is not needed, please delete it. Generally, I do not see the necessity of the section 4.1. It better fits for the introduction. Fig10: I did not find any reference to this figure in the text. L379: What do you mean with applicable conditions? L383: what do you mean by stating that the rain-fed agriculture depends on groundwater? If it is rain-fed, it is not irrigated. Thus, groundwater is not used. If you mean the moisture, which is stored in the soil and used by the plants, it is the green water and not the blue water. Please revise this sentence. L429-433: You state that the method you developed also applies for the rain-fed agriculture. This is correct, because then you just exclude the irrigation parameter from the SWAT model. Nevertheless, this is a commonly used method and I do not see any novelty of your method here. For this reason, I recommend to delete this section. L456-457: You already stated in the L45, that blue water has the largest part of the total WF. Please delete this sentence.

---

## Referee Comment (RC2) · Anonymous Referee #2 · 9 May 2018

Referee comment on

Title: An improved method for calculating regional crop water footprint based on hydrological process analysis

Author(s): Xiao-Bo Luan et al.

MS No.: hess-2018-125

MS Type: Research article

Special Issue: Integration of Earth observations and models for global water resource assessment

[Figure]

**SUMMARY AND GENERAL COMMENTS**

The paper presents a semi-distributed approach to model effective water resource requirements in crop production in terms of the volume of water used per unit crop production. The approach differentiates between green and blue water sources and puts emphasis on conveyance losses of irrigation water. Modelling of the water cycle is based on SWAT, while conveyance losses between the water inlet of the irrigation scheme and the field are modelled depending on the location according to a new approach that, apparently, has not been published before. The novel contribution to the field of science by this study is limited to the location-dependent modelling of conveyance losses, which can potentially have significant effect on crop water footprint calculations. Unfortunately, the derivation of the approach is neither explained in much detail nor is its validity tested against measured data. Overall, the presentation of the theoretical background, methods and results is rather poor and, at least partly, hard to understand. The language is unprecise and redundant in major parts of the paper. It leaves room for interpretation (eg lines 64-66) and numerous sentences/paragraphs are unintelligible (e.g. lines 86-87,90-93, 104-105, 207-209). I am not a native English speaker but I feel the text needs revision with regards to pure language issues (grammar, mode of expression). The paper does not provide a critical discussion of the approach and the results. In particular, uncertainties of inputs and results are hardly addressed. Major parts of the discussion section basically repeat the contents of the introduction. The conclusions section is basically a summary of the results and the few conclusions made are trivial. The title does not match the content of the manuscript (see comment on the term "water footprint" below).

**DETAILED COMMENTS ON SUBSTANTIAL SHORTCOMINGS OF THE MANUSCRIPT**

The authors refer to the water resource requirements of crop production as "water footprint", which is inappropriate two reasons. Firstly, indirect water uses, an important aspect of a footprint indicator, are not considered in the study. Secondly, the paper lacks

a clear definition of the system (consumer or producer) that causes the footprint. The paper presents water resource requirements for the production of three different crops (m$^3$ water use/t of crop production, referred to as "water footprint") in subbasins of the Hetao Irrigation District (HID). Obviously, the "water footprint" is defined for a producer. It is not stated whether the footprint figures are calculated for (a) a single producer, i.e., the aggregate of "farms" growing a single crop type in the HID, or (b) many different producers, i.e., the aggregates of farms growing that crop within individual subbasins. However, this is important in order to understand the results correctly. In case (a) the volume of water used to produce xi tonnes of crop in subbasin i needs to be related to the total crop production in HID (X). If ri is the water resource requirement in sub-basin i, the water footprint of the HID-wide crop production in subbasin i calculates as Fi=xi/X*ri. In contrast, the water footprint of subbasin-wide crop production (case (b)) in subbasin i is given as F'i=ri. Note that in case (b), the "water footprint" indicator is no longer geographically explicit, another important aspect of the water footprint, as the subbasins are the smallest geographical units presented. The range of results shown in the maps implies that the water footprint is defined according to case (b). However, water resource requirements for crop production are intrinsic properties of the irrigation system in each subbasin and are independent of the actual allocation of crop produc-tion. Hence, the study is not a footprint analysis but, simply, an analysis of resource requirements (comparable to a potential analysis). However, the representativeness of the results is questionable due to methodological limitations. Subbasins are sub-divided into hydrological response units (HRU) based on land use (supposedly land use=crop type) and soil type. Although it is not stated explicitly, one must assume that the results on HRU-level, based on the actual pattern of crop allocation and irrigation timing/quotas, are aggregated to subbasin-level (aggregation method not specified). This way, the results are only representative for potentially small parts of a subbasin, i.e., one or more HRUs within a subbasin under the given crop, as the conditions (soil type, canal losses, etc.) may be different in the remaining parts of the subbasin. The reader cannot judge the related uncertainties as the actual patterns of crop allocation

and soil types are not shown. The description of the methods to calculate the "water footprint" is difficult to understand. As the system boundaries are not defined precisely, the reader is forced to examine several possible system boundaries in order to judge whether the equations 6-9 are likely to be correct. For instance, it depends on the system boundary whether field discharge (Qd) is actually consumption, i.e. it is a flow out of the system (to another basin or the sea), or returns to system itself. As the authors stress that the approach is regional-scale, a certain share in field discharge is likely a return flow, which would invalidate equation 7, which defines field discharge as water consumption. Equations 6-9 use a set of variables that are calculated for two different scenarios (s1=with irrigation, s2=without irrigation) but the notation is ambiguous as the scenario is not clearly indicated in the equations except for for ET (index s1 or s2). It might be considered obvious that canal losses (Qc) and ET of field irrigation (Qf) is only defined for the scenario with irrigation (s1). (Note, those variables can also be defined for s2, though with a value of zero.) However, capillary rise of groundwater (Qg) and field discharge (Qf) definitely can have non-zero values for s2. Hence, it must be indicated from which scenario the values are taken. Qg must not be added in eq 7. Although Qg is per definition blue water, it simple changes soil moisture. The share of Qg that is consumed is already included in Qf+Qd. As I understand, canal losses in eq 7-8 are informed by the modelling approach represented by eq 10-15 but it remains unclear which of the variables mentioned in eq 10-15 are actually used and how. The notation of eq. 10-15 is confusing as I suspect most readers are familiar with a notation where n is the total number of elements and i is a running index. Here, it is used the other way around, which is not wrong but makes it more difficult to understand. The section on calibration and validation of the model is wordy and interrupts the description of the modelling approach. For instance, the $R^2$ metric is widely used and there is no need to show the formula. If equations 2-4 are considered necessary, the notation should be corrected as the index i is missing in numerous terms.

CONCLUSIONS

Given the shortcomings addressed above, the quality of the manuscript is, in my opinion, not acceptable for publication, although the underlying material fits the scope of the journal and might be worth publishing. Due to missing definitions and precise description of the methods, I can hardly judge the validity of the work. I think the necessary revisions are too extensive to be done within a peer-review process. Apart from this, addressing all the issues where I see the need for revision in this reviewer comment would be an unreasonable effort. Therefore, my recommendation is to reject the paper.

Please also note the supplement to this comment:
https://www.hydrol-earth-syst-sci-discuss.net/hess-2018-125/hess-2018-125-RC2-supplement.pdf

―――――――――――――――――

---

## Author Comment (AC1) · 4 Jul 2018

<Manuscript number: HESS-2018-125> Dear Editors and Reviewers: Thank you for your letter and for the reviewers' comments concerning our manuscript "An improved method for calculating regional crop water footprint based on hydrological process analysis". We appreciate your comments and constructive suggestions very much, and they were valuable for improving the quality of our manuscript. We have revised the manuscript in detail according to the editor and reviewers' comments. We hope that these modifications, based on your suggestions and the reviewers' comments, will raise the quality of our manuscript to meet the publication standards of Hydrology and

[Figure]

Earth System Sciences. The revised portions are marked in red in the paper. The main corrections in the paper and the responses to the reviewer's comments are as follows:

Anonymous Referee #1

In this manuscript the authors enhance the Water Footprint method to a regional scale. For that purpose, the water losses in the irrigation water distribution system are included into the WF (blue water). A procedure for the quantification of water losses in the canal system is introduced. The topic addressed in this manuscript is relevant and has certain novelty for the water footprint assessment. Nevertheless, several minor and major aspects need to be improved. The overall calculation approach needs to be checked, since some calculation steps are contradictory to the common water footprint method. The grammar is very poor throughout the whole paper and has to be improved. Response: Thank you for your recognition of the innovation of the paper. In the revised manuscript, we describe the difference of the calculation method of water footprint between this study and the traditional method, as well as the innovation of this study. At the same time, we improve the language of the thesis (The paper was edited by Elsevier Language Editing Services).

General comments: 1. Comment: Water loss from the canal system and application on the fields is calculated as water consumption. This is not correct, because leached water recharges the local groundwater aquifers and, thus, contributes to the water availability. In that term, it is not consumed. Therefore, water consumption equals the actual evapotranspiration (for the agricultural production), while water losses (leaching from the fields) is subtracted from the water used for the irrigation. Therefore, I do not understand why you calculate it as water footprint. I can understand if you state that water is recharged in another aquifer, however the groundwater aquifers are connected. Please consider that issue with the water losses and the spatial aspect of water losses. The second scenario is not plausible, because, as stated later on in this paper, 75% of the water input comes from the irrigation, that is why this is not a red-fed agriculture and stopping all irrigation will lead to a total yield loss. Therefore,

this scenario cannot exist in reality. If you want to develop a scenario, please consider, for example, deficit irrigation. Nevertheless, I do not see the necessity of the second scenario for your paper. The methodology for calculating WF on the field level (so, for the rain-fed agriculture) already exists and is broadly applied. Therefore, there is no novelty in your approach concerning that aspect. Furthermore, the results of the second scenario are missing in the section Results. Response: During the agricultural production process, especially in irrigated agriculture, the use of water by crops has undergone the following processes: first, water is diverted from the water source (river, lake or groundwater) and then transported to the field through irrigation canals or water pipelines, where crops are irrigated through various irrigation methods. In the process of water diversion, part of the water will be lost, one is the evaporation and leakage in the canal or pipeline, the other is the evapotranspiration, runoff and leakage in the field irrigation. These losses are paid to make sure the crop production. In this study, the loss of irrigation water in the course of water transportation was included in the calculation of the water footprint of crop production. The water lost in the process of irrigation water transportation is contained in the local planned water intake, which leads to the local water diversion being higher than the actual net irrigation amount of the crop. At the same time, the water distribution facility needs to be built during the transportation process, which consumes a large amount of labor, capital and technology. Therefore, from the perspective of agricultural water management, the loss of irrigation water should also be included in the calculation of the water footprint of crop production. In addition, although the water lost in the canal system and the field leakage can be redeveloped and utilized, it also needs to consume a large number of labor, capital and technology. Meanwhile, the redeveloped water can be considered as an additional source of new irrigation water, and local water use plans need to be adjusted, which can also affect local water resources management. Therefore, for agricultural production, the water consumption in crop production is calculated based on the crop water consumption, and the above-mentioned loss of crop-related water is calculated, which is conducive to the local water management department to allocate and

manage the regional water resources. In this study, the Scenario 2 was set to calculate the consumption of green water. In this study area (HID), because of less rainfall, the effective precipitation formed by precipitation is all used for crop growth. Therefore, the consumption of green water for crops is equal to the effective precipitation, which means that green water is reflected by calculating the effective precipitation stored in soil by SWAT model. We modified the formula, as follows: (6) Where $Wg$ is the green water consumption during the crop growth period (m3), PRECIPs2 is the precipitation during the crop growth period in Scenario 2 (m3), SUPQs2 is the surface runoff during the crop growth period in Scenario 2 (m3), LATQs2 is the soil lateral flow during the crop growth period in Scenario 2 (m3).

2. Comment: Regarding the term ET, it would be good to distinguish between the potential and actual ET. I would appreciate if you specify it in the text. Response: Thank you for your comment. We have modified the description in the revised manuscript as your suggestion. The modified parts are as follows: The first is the crop water requirement method. This method simulates the actual evapotranspiration (ET) of crops under optimal conditions with the potential ET calculated by the Penman-Monteith Equation (Allen et al., 1998). (Page 4, line 74, 75.) The green water consumption is the smaller value of total crop actual ET and effective precipitation. The blue water consumption is obtained through the difference between the total crops actual ET and effective precipitation. (Page 4, line 76-79.) These methods can simulate actual ET throughout the crop growing period according to the soil water balance under optimal or suboptimal conditions. (Page 5, line 96.) The green water consumption is equal to the total actual ET minus blue water. (Page 5, line 99.)

3. Comment: I would appreciate if you also address the importance of efficiency of the water distribution systems in the Discussion Response: Thank you for your suggestion. We have added the importance of efficiency of the water distribution systems in the Discussion section. The modified parts are as follows: Page 24, line 445-452. 4.4 The influence on efficiency of irrigation system The efficiency of irrigation system is

affected by the way of water transportation, the condition of canal system, the irrigation technology and so on. Therefore, the water use efficiency of the regional irrigation system can be improved by changing the water delivery method (from the channel to the pipeline) and the irrigation method (such as dropper, sprinkler and other advanced irrigation technologies). For the study area, the results show that more than half of the water resources were lost during the process of canal water transport and irrigation. Therefore, the use of anti-seepage measures to reduce the leakage of canal systems, while the use of advanced irrigation technology to reduce the amount of irrigation water is conducive to reduce the water footprint of crop production of the region.

4. Comment: Please improve the grammar throughout the whole paper. Your sentences are sometimes built in a strange way and are difficult to understand. I recommend to ask a native speaker to review the paper. Response: Thank you for your suggestion. We have carefully revised the language of the paper and invited native speaker to polish the manuscript (The paper was edited by Elsevier Language Editing Services).

Specific comments and technical corrections: 1. Comment: L21: effectively managing -> effectively manage. Response: Thank you for your comments. We have modified this description in the revised manuscript as suggested. The modified parts are as follows: Page 2, line 21. With the shortage of water resources, assessing the water use efficiency is crucial to effectively manage agricultural water resources.

2. Comment: L22: It is not correct to state that WF is a new method; it is used in the scientific community since decades. I would recommend deleting this statement. Abstract: please address that you are quantifying the WF in terms of blue and green water, because there are other methods, for example impact assessment using the AWARE, WSI and WAVE models, but also the grey water footprint. Response: Thank you for your comments. We have modified this description in the revised manuscript as suggested. The modified parts are as follows: Page 2, line 22. The water footprint is an improved index for water use evaluation, and it can reflect the quantity and types of

water usage during crop growth. This study aims to establish a method for calculating the region-scale water footprint of crop production based on hydrological processes, and the water footprint is quantified in terms of blue and green water.

3. Comment: L26: do you mean with the term "underground water"? Groundwater use? Please use another term. Response: Thank you for your comments. We have modified this description in the revised manuscript as suggested. The modified parts are as follows: Page 2, line 26. This method analyzes the water-use process during the growth of crops, which includes irrigation, precipitation, groundwater, evapotranspiration, and drainage, and it ensures a more credible evaluation of water use.

4. Comment: L34: further -> further away. Response: Thank you for your comments. We have modified this description in the revised manuscript as suggested. The modified parts are as follows: Page 2, line 35. The spatial distribution pattern of the green, blue and total water footprint for the three crops demonstrated that higher values occurred in the eastern part of the HID, which had more precipitation and was further away from the irrigating gate.

5. Comment: L44: to ensuring -> to ensure. Response: Thank you for your comments. We have modified this description in the revised manuscript as suggested. The modified parts are as follows: Page 3, line 45. In China, 63% of all water is used for agricultural production each year, and the area of irrigated farmland is 39.6% of the total arable land. Irrigation is the key to ensure agricultural production (NBSC, 2016).

6. Comment: L54: utilization of crop production -> for crop production. Response: Thank you for your comments. We have modified this description in the revised manuscript as suggested. The modified parts are as follows: Page 3, line 55. Strengthening agricultural water management and improving water use efficiency are significant aspects of handling water scarcity, and a reasonable evaluation of the water resource for crop production is the premise for developing an agricultural water management plan and implementing water saving measures.
7. Comment: L57: What do you mean with "reduce the negative impact of the re-
duction of available agricultural water"? Please check and correct this sentence. Re-
sponse: Thank you for your comments. We have modified this sentence in the revised
manuscript as suggested. The modified parts are as follows: Page 3, line 57. There-
fore, how to precisely evaluate the effective utilization ratio of current agricultural water
use, improve the utilization efficiency, and reduce the negative impact of the reduction
of available agricultural water on agricultural production, is an important issue that all
countries need to address globally, this is also of vital importance for ensuring food
production and reducing the pressure on water resources.

8. Comment: L58: Globally -> globally. Response: Thank you for your comments. We
have modified this description in the revised manuscript as suggested. The modified
parts are as follows: Page 3, line 59. Therefore, how to precisely evaluate the effective
utilization ratio of current agricultural water use, improve the utilization efficiency, and
reduce the negative impact of the reduction of available agricultural water on agricul-
tural production, is an important issue that all countries need to address globally, this
is also of vital importance for ensuring food production and reducing the pressure on
water resources.

9. Comment: L62: Please address and explain here the terms green, blue and grey
water instead of the term "types of resources". Response: Thank you for your com-
ments. We have modified this sentence in the revised manuscript as suggested. The
modified parts are as follows: Page 4, line 63. It reflects the amount of water, the
green, blue and grey water that are consumed (Hoekstra, 2011).

10. Comment: L72: This data is not necessarily provided by USDA, it is available in
other sources. Of course, you used this data source for your case study, but here it is
better to delete the reference to the USDA. Response: Thank you for your comments.
We have modified this reference in the revised manuscript as suggested. The modified
parts are as follows: Page 4, line 76. This method simulates the evapotranspiration
(ET) of crops under optimal conditions with the ET calculated by the Penman-Monteith

Equation (Allen et al., 1998) and the effective precipitation calculation method refer to Doll and Siebert (2002). Reference: Doll, P., Siebert, S. (2002). Global modeling of irrigation water requirements. Water Resources Research, 38(4):1037-1048.

11. Comment: L76: Please indicate, that the WF is calculated per kg or ton of crop. Response: Thank you for your comments. We have modified this sentence in the revised manuscript as suggested. The modified parts are as follows: Page 4, line 79. Finally, when combined with crop yields, the crop blue and green water footprint (m3/t) can be calculated.

12. Comment: L68-81 – please insert the equations for the calculation of green and blue WF. Response: Thank you for your comments. We have added the equations in the revised manuscript as suggested. The equation (2) is the crop water requirement method, the equation (3) is the irrigation schedule method. The modified parts are as follows: Page 5, line 83-95. These two methods formulas are as follow, the equation (2) is the crop water requirement method, and the equation (3) is the irrigation schedule method. (1) (2) (3) where CWUgreen is the green component in crop water use, CWUblue is the blue component in crop water use, ETgreen is the green water evapotranspired, ETblue is the blue water evapotranspired, Y is the crop yield, ETc is the crop evapotranspiration, Peff is the efiective rainfall, Kc is the crop coefficient, ET0 is the reference evapotranspiration, IRRt is the total net irrigation, IRRa is the actual irrigation requirement, ETa is the adjusted crop evapotranspiration, Ks is the soil water stress coefficient, describes the efiect of water stress on crop transpiration, For soil water limiting conditions, Ks < 1; when there is no soil water stress, Ks = 1. These equations are based on CROPWAT model.

13. Comment: L80 – Please explain the terms net irrigation water and the actual irrigation water requirement. Response: Thank you for your comments. We have modified this sentence in the revised manuscript as suggested. The Net irrigation water is the amount of water actually irrigated to the field. The net irrigation water requirement is the actual irrigation amount needed in the field. The modified parts are

as follows: Page 5, line 97, 98. The blue water consumption is the smaller value of net irrigation water and the net irrigation water requirement.

14. Comment: L89: I agree that the existing methods do not consider the water losses in canals, but the water leached from the fields (if I correctly understand the term "drainage water") is actually considered. Therefore, please revise this statement. Response: Thank you for your comments. We have modified this sentence in the revised manuscript as suggested. The modified parts are as follows: Page 6, line 105-107. These methods calculated the field-scale water footprint with net irrigation water considered as irrigation water, and without considering water loss during transport, which definitely serve for crop growth.

15. Comment: L95: The term "irrigation quota" is not clear. Do you mean the irrigation demand? Please add the definition and probably also use another term throughout the whole paper. Response: Thank you for your comments. We have modified this sentence in the revised manuscript as suggested. The irrigation quota is the irrigation demand. The modified parts are as follows: Page 6, line 114. Second, the irrigation data in these methods are simulation values and not based on the actual irrigation time and irrigation quota (the amount of water demanded for crop irrigation);

16. Comment: L96: I would not define it as incorrect, rather not complete. Response: Thank you for your comments. We have modified this sentence in the revised manuscript as suggested. The modified parts are as follows: Page 6, line 115. Therefore, these data cannot reflect the real situation of the local water usage due to the incomplete simulation data.

17. Comment: L97: that is not true, precipitation and irrigation water are distinguished, that is why it is possible to distinguish between the green and blue water. The irrigation water is calculated as the amount of water to meet the part of the crop water requirement, which cannot be fulfilled by the precipitation. Concerning the groundwater, you are correct: surface water and groundwater are evaluated together for the blue water

calculation. Response: Thank you for your comments. This description is indeed inaccurate. According to your suggestion, we have deleted this sentence. The modified parts are as follows: Page 6, line 116, 117. The traditional method does not completely analyse the water footprint components of water resources in the process of water diversion, water transfer, irrigation and drainage.

18. Comment: L99-101: it is not clear how it is different from the other limitations you mentioned above. Response: Thank you for your comments. We have modified this sentence in the revised manuscript as suggested. The modified parts are as follows: Page 6, line 118, 119. The method that Sun et al. (2013b) used also had these limitations which mentioned above.

19. Comment: L106-107 – this a repetition. Response: Thank you for your comments. According to your suggestion, we have deleted this sentence.

20. Comment: L109 – 115 – the sentences are too long and difficult to understand. Please rewrite this section. Response: Thank you for your comments. We have modified these sentences in the revised manuscript as suggested. The modified parts are as follows: Page 7, line 126-133. This method simulated the hydrological cycle of the region based on a physical hydrological model (SWAT). Based on the method, this study analyzed the water input and output during crop production, and calculated the water consumption in crop growth, field drainage and water loss during canal water transport. Combined with crop yields, the water footprint of crop production at the regional scale was quantified. This method will provide comprehensive information for the analysis of water consumption during crop production process and improve the spatial resolution of the regional distribution of water footprint of crop production.

21. Comment: L121-122: Since it has monsoon climate, the annual average values for precipitation and ET do not provide any valuable data. Consider providing average values for the dry season and monsoon season. Response: Thank you for your comments. We have modified this description in the revised manuscript as suggested. The

modified parts are as follows: Page 7, line 139-141. The average monthly precipitation is 37.5 mm (May to September), 3.4 mm (October to next year April), and the average monthly potential evaporation is 290.6 mm (April to September), 77.2 mm (October to next year March).

22. Comment: Fig 1: Please use another word for the areas (Dengkou etc), e.g. districts in the legend, because the term "country" is misleading. Response: Thank you for your comments. We have modified this description in the revised manuscript as suggested. The Hetao Irrigation District mainly consists of 5 counties, namely, Dengkou, Hangjinhouqi, Linhe, Wuyuan, and Wulateqianqi. The modified parts are as follows: Page 8, line 147.

Fig. 1. Location of the Hetao Irrigation District (HID) in China

23. Comment: L125: Could you please explain more the differences between canals and ditches? Response: Thank you for your comments. The canal is the engineering measure that the water is transported from the water source to the field. The drainage ditch is the engineering measure to guide the surplus water from the field into the river or lake.

24. Comment: L170-172: I do not see a benefit of writing the formulas 2, 3 and 4 into the text. I recommend deleting them. Response: Thank you for your comments. According to your suggestion, we have deleted these formulas (R2, NSE, and PBIAS).

25. Comment: L175-176: The explanation of the R2 is not needed, since it is a very common parameter in statistics. Response: Thank you for your comments. According to your suggestion, we have deleted the explanation of the R2.

26. Comment: L166-189: This section includes information on the uncertainty analysis, which is a bit confusing. It distracts the reader from the actual content of the paper. It will be better to put this data into the supplementary material and only mention that the results are satisfactory. Response: Thank you for your comments. According to your

suggestion, we have put this data into the supplementary material.

27. Comment: L203: What exactly do you mean with the term "irrigation water in the fields"? Surface water from the irrigation channels? Please use another term, otherwise it is not clear what you mean. Response: Thank you for your comments. We have modified this description in the revised manuscript as suggested. The modified parts are as follows: Page 11, line 201. In scenario 1 (S1), crop water consumption was derived from precipitation and irrigation water (irrigation systems and irrigation quotas are based on local irrigation methods), i.e., the actual situation of crop water use.

28. Comment: L205: I do not understand how this scenario should work in reality. If the fields are irrigated, this means, that there is not enough precipitation to meet the crop water requirement. That would mean that in your scenario 2 the crops do not have enough water. So, do you also consider the yield loss? Because you cannot get the same yield under such different conditions. Response: In this study, the Scenario 2 was set to calculate the consumption of green water. In this study area (HID), because of less rainfall, the effective precipitation formed by precipitation is all used for crop growth. Therefore, the consumption of green water for crops is equal to the effective precipitation, which means that green water is reflected by calculating the effective precipitation stored in soil by SWAT model. We modified the formula, as follows:

Where Wg is the green water consumption during the crop growth period (m3), PRE-CIPs2 is the precipitation during the crop growth period in Scenario 2 (m3), SUPQs2 is the surface runoff during the crop growth period in Scenario 2 (m3), LATQs2 is the soil lateral flow during the crop growth period in Scenario 2 (m3).

29. Comment: Equations 5-9 – please check the formatting. Eq.5 – this equation is obvious, you can describe it in the text (total WF is the sum of the green and blue WF), but you don't need an equation for that. Eq. 6- please provide more explanation. It is not clear why the groundwater, which raises to the soil plow layer, is included into the green WF calculation. Eq.7: Qd – same as for the water loss in the canal – this is not a

loss because the water recharges the GW aquifer, so actually the equation should be Qf-Qd. Again, including Qg into the water consumption is not clear. Response: Thank you for your comments. We have modified these formulas in the revised manuscript as suggested. In this study, the field discharge (Qd) flow out of the Hetao irrigation district, those water could not been used again, so, the field discharge (Qd) is a part of blue water consumption. The modified parts are as follows: Page 12, 13, line 209-226. (5) (6) (7) (8) (9) (10) where WF is the water footprint of crop production (m3/t), WFg is the green footprint (m3/t), WFb is the blue water footprint (m3/t), Wg is the green water consumption during the crop growth period (m3), Wb is the blue water consumption during the crop growth period (m3), Y is the crop yield (t), PRECIPs2 is the precipitation during the crop growth period in Scenario 2 (m3), SUPQs2 is the surface runoff during the crop growth period in Scenario 2 (m3), LATQs2 is the soil lateral flow during the crop growth period in Scenario 2 (m3), Qc is the amount of water loss in the canal system (m3), Qf is the actual ET of field irrigation water (m3), Qd is the field discharge (m3), It,s1 is the amount of total irrigation water diversion in Scenario 1 (m3), and If,s1 is the actual amount of water irrigated in the field in Scenario 1 (m3). ks1 is the effective utilization coefficient of canal water in Scenario 1 (Obtained from the local Water resources management department), ETs1 is the crop actual ET during the crop growth period in Scenario 1 (m3), WYLDs1 is the total amount of water leaving the HRU in Scenario 1 (m3). The data of parameters PRECIPs2, SUPQs2, LATQs2, It,s1, ETs1, WYLDs1 were obtained from the SWAT model.

30. Comment: Fig4 – please include explanation of Acronyms and indices into the name of the table. Could you also show the part A and part B on the figure? Response: Thank you for your comments. We have added explanation of acronyms and indices in the revised manuscript as suggested. The figure of Part A and Part B are the intermediate process of the calculation process. We input the data of Part A and Part B into ArcGIS to obtain figure a and figure b, then add the two pictures to obtain the figure of final result, which is Figure 6(Canal water loss). Because the two figures are intermediate process figure, it could not been displayed in the paper. The modified

parts are as follows: Page 15, 16, line 272-280.

Fig. 4. Model for calculation of water loss in canal system Note: Sjn is the area of each sections in the jth main canal, Wjn is the water loss per unit area of the section of the jth main canal in Part A, Qji is the actual amount of water loss per unit area of the i section of the jth main canal, Sj is the area controlled by the jth main canal, kj is the coefficient of the water distribution from the general main canal to the jth main canal, Qj is the water loss per unit area of the jth main canal in Part B, kgc is the water conveyance efficiency of the general main canal, kmc is the water conveyance efficiency of the main canal, j is the number of the main canal, i is the number of the equidistance sections in the jth main canal.

31. Comment: Eq. 10-15 – check formatting of the numeration of the equations. It is difficult to follow the equations. It would help to understand, if you split them into calculation for the part A and part B and include the explanations of the indices directly after each equation. More description of the whole calculation path is needed to follow the calculation procedure. Response: Thank you for your comments. We have modified this description in the revised manuscript as suggested. The modified parts are as follows: Page 13-15, line 228-271 Water transfer loss is a kind of water loss in the process of channel water delivery, and it is an important part of blue water consumption in crop production. For a piece of cultivated land, the water loss during the process of the crop production includes the loss of water from the water source to the field flowing through the canal system. In the Hetao Irrigation District, irrigation canal is composed of seven grades (general main canal, main canal, sub-main canal, branch canals, lateral canals, field canals, and sub-lateral canals). Because of the complex distribution of canal system and the lack of hydrological data in irrigation districts (the lack of effective utilization coefficient of canal water below the main canal). Therefore, in calculating the water loss of canal system during crop production process, we generalized Hetao Irrigation District into a model similar to the histogram (Fig. 4). We divide the total water loss of canal system into two parts. Part A is the loss of the main canal and canal,

and Part B is the loss of the remaining canal system (the water loss of the sub-main canal and its sub-channels at all levels). The calculation of water loss in part A is as follows: first, the water loss of each section is calculated by dividing the main canal into equal distances (10 km). Then the water transfer loss of each section of the canal is allocated to each field downstream [Equation 10], thereby obtaining the water transfer loss in the crop production process on the field block. Therefore, the actual water loss caused by irrigation in a field is the sum of the water loss of the transfer canal and the canal in the upstream. We assign the actual water loss of the field by irrigation (Qji, formula 11) to the midpoint of the each section, and use Kriging interpolation in ArcGIS to obtain the water loss distribution map of the figure a (Part A). Due to the lack of the effective utilization coefficient of canal water and the distribution map of the canals at all levels and below, the calculation process of the water loss in Part B is as follows: the remaining canal loss in each irrigation canal is divided by the main canal irrigation and the unit area loss of the canal control area is obtained. Then, the amount of water loss per unit area within the control range of each main canal in the irrigation area (Qj, formula 15) is obtained, and the data is brought into ArcGIS for the water loss distribution map of figure b (Part B). Finally, the figure a and the figure b are superimposed and calculated in the ArcGIS using the map algebra module of the spatial analysis tool to obtain the water loss distribution map of the canal system in HID.The formulas are as follows: (11) (12) (13) (14) (15) (16) where Qji is the actual amount of water loss per unit area of the i section of the jth main canal in Part A( m3/ha), Wjn is the water loss per unit area of the section of the jth main canal in Part A (m3/ha), j is the number of the main canal, i is the number of the equidistance sections in the jth main canal, n is the total number of the sections in the jth main canal, m is the total number of the main canals, WA is the amount of water loss in Part A (m3), kj is the coefficient of the water distribution from the general main canal to the jth main canal, Sjn is the area of each sections in the jth main canal (ha), It,s1 is the amount of total irrigation water diversion in Scenario 1(m3), kgc is the water conveyance efficiency of the general main canal, kmc is the water conveyance efficiency of the main canal, Sj is the area controlled by

the jth main canal (ha), Qj is the water loss per unit area of the jth main canal in Part B (m3/ha), WB is the amount of water loss in Part B (m3), and Qc is the amount of water loss in the canal system (m3).

32. Comment: L253: Do you mean "sections"? Response: Thank you for your comments. We have modified this description in the revised manuscript as suggested. The modified parts are as follows: Page 15, line 264. i is the number of the equidistance sections of the jth main canal.

33. Comment: L255: Qn is the actual amount.... Response: Thank you for your comments. We have modified this description in the revised manuscript as suggested. The modified parts are as follows: Page 15, line 262. Qji is the actual amount of water loss per unit area of the ith section of the jth main canal in Part A (m3/ha).

34. Comment: L270: I do not understand for which parameters these rates are. Precipitation, irrigation on field and canal loss? Response: Thank you for your comments. We have modified this description in the revised manuscript as suggested. The modified parts are as follows: Page 16, line 289-293. Precipitation and irrigation are the water input items in the process of crop production, and the canal water loss, field actual ET and field drainage are the water output items. For water input, precipitation and irrigation accounted for 25.1% and 74.9%, respectively. For water output, channel water loss, field actual ET and field drainage accounted for 47.9%, 41.8% and 10.3%, respectively.

35. Comment: L290: Since you didn't consider the groundwater irrigation, please indicate here, that the blue WF includes only the surface water irrigation. Response: Thank you for your comments. We have modified this description in the revised manuscript as suggested. The modified parts are as follows: Page 17, line 313. Blue water is the surface water used for crop growth in this study.

36. Comment: L299-300: I do not see this correlation in your results. Actually, there is rather more field discharge by larger ET, if I understand your figures correctly. Please

check this data. It is also interesting whether there are same irrigation techniques applied on the whole study area. Because water loss on the field depends on the irrigation method applied. Please address this aspect. Response: For the field, actual ET and drainage are the mainly water output items. So the ET increase will lead to the decrease of the field drainage when the amount of water in the field soil is certain. In this study, due to the large area of irrigation areas, and farmland were planted by a large number of farmers, its irrigation time, irrigation water will have some differences. In order to reduce the complexity of the study, irrigation time and irrigation water were set to the same parameters in the entire Hetao irrigation district in the SWAT model.

37. Comment: L302: This sentence is a repetition, please delete. Response: Thank you for your comments. We have deleted this sentence in the revised manuscript as suggested.

38. Comment: L322-325 and L310-311: These statements are obvious, because these three crops have different crop water requirement. Please consider deleting these sentences. Response: Thank you for your comments. We have deleted these sentences in the revised manuscript as suggested.

39. Comment: L345: Do you mean "water footprints on the regional scale and field scale"? Response: Thank you for your comments. We have modified this description in the revised manuscript as suggested. The modified parts are as follows: Page 20, line 365. The region scale and field-scale methods for calculating crop production water footprints

40. Comment: L346: Method for. Response: Thank you for your comments. We have modified this description in the revised manuscript as suggested. The modified parts are as follows: Page 20, line 366. In this paper, the calculation method for calculating crop production water footprints is divided into the field scale and region scale.

41. Comment: L350-353: I understand what you want to say, but this sentence is misleading, because you firstly states, that the studies are on national, regional etc scale,

but then says that the studies are on the field scale. Please change the rephrase to make it more understandable. Response: Thank you for your comments. We have modified this description in the revised manuscript as suggested. The modified parts are as follows: Page 4, line 69-73. Many scholars have quantified various levels of crop production water footprints, such as a global level (Mekonnen and Hoekstra, 2011), a national level, such as Europe (Vanham and Bidoglio, 2013) and China (Zhao, 2009), and a regional level, such as Beijing (Sun, 2013a), Cremona province (Bocchiola, 2015) and Hetao (Luan et al., 2018).

42. Comment: L350-362: There is too much text explaining the methods. I recommend deleting Response: Thank you for your comments. We have inserted these sentences into the introduction in the revised manuscript as suggested in comment 43. The modified parts are as follows: Page 4, 5, line 73-83. The first is the crop water requirement method (Cao et al., 2014; Sun et al., 2013c). This method simulates the actual evapotranspiration (ET) of crops under optimal conditions with the potential ET calculated by the Penman-Monteith Equation (Allen et al., 1998) and the effective precipitation calculation refer to Doll and Siebert (2002). The green water consumption is the smaller value of total crop actual ET and effective precipitation. The blue water consumption is obtained through the difference between the total crops actual ET and effective precipitation. Finally, when combined with crop yields, the crop blue and green water footprint (m3/t) can be calculated. The second is the irrigation schedule method. This method is based on an empirical formula model such as the CROPWAT model (FAO, 2010; Mekonnen and Hoekstra, 2011) CropSyst (Bocchiola et al., 2013), the EPIC model (Williams et al., 1989; Shi et al., 2017), the GEPIC model (Liu et al., 2007), and the AQUACROP model (Pasquale et al., 2009; Chukalla, 2015; Zhuo, 2016).

43. Comment: L353-360 or insert them into the introduction and refer to it here. Response: Thank you for your comments. We have insert these sentences into the introduction in the revised manuscript as suggested The modified parts are as follows: Page 4, 5, line 69-83. Currently, based on two mainly methods proposed by Hoekstra

et al. (2011), many scholars have quantified various levels of crop production water footprints, such as a global level (Mekonnen and Hoekstra, 2011), a national level, such as Europe (Vanham and Bidoglio, 2013) and China (Zhao, 2009), and a regional level, such as Beijing (Sun, 2013a), Cremona province (Bocchiola, 2015) and Hetao (Luan et al., 2018). The is the crop water requirement method (Cao et al., 2014; Sun et al., 2013c). This method simulates the actual evapotranspiration (ET) of crops under optimal conditions with the potential ET calculated by the Penman-Monteith Equation (Allen et al., 1998) and the effective precipitation calculation refer to Doll and Siebert (2002). The green water consumption is the smaller value of total crop actual ET and effective precipitation. The blue water consumption is obtained through the difference between the total crops actual ET and effective precipitation. Finally, when combined with crop yields, the crop blue and green water footprint (m3/t) can be calculated. The second is the irrigation schedule method. This method is based on an empirical formula model such as the CROPWAT model (FAO, 2010; Mekonnen and Hoekstra, 2011) CropSyst (Bocchiola et al., 2013), the EPIC model (Williams et al., 1989; Shi et al., 2017), the GEPIC model (Liu et al., 2007), and the AQUACROP model (Pasquale et al., 2009; Chukalla, 2015; Zhuo, 2016).

44. Comment: L366-370 –This text is not needed, please delete it. Generally, I do not see the necessity of the section 4.1. It better fits for the introduction. Fig10: I did not find any reference to this figure in the text. Response: Thank you for your comments. We have deleted these sentences in the revised manuscript as suggested and have added the reference of Fig. 10. The modified parts are as follows: Page 20, 21, line 373,374. Fig. 10 is the calculation range of the regional scale and field scale method of crop production water footprint.

45. Comment: L379: What do you mean with applicable conditions? Response: Thank you for your comments. We have modified this description in the revised manuscript. The applicable condition is the calculation boundary of the two methods. The modified parts are as follows: Page 21, line 387-389. The calculation boundary of the two

methods of calculating water footprints are different, the calculation of the green water footprint is the same, whereas the calculations of the blue water footprint are different.

46. Comment: L383: what do you mean by stating that the rain-fed agriculture depends on groundwater? If it is rain-fed, it is not irrigated. Thus, groundwater is not used. If you mean the moisture, which is stored in the soil and used by the plants, it is the green water and not the blue water. Please revise this sentence. Response: Thank you for your comments. We have modified this description in the revised manuscript as suggested. The modified parts are as follows: Page 21, line 389-391. The rainfed agriculture depends on precipitation (green water), and the water consumption mainly includes actual ET.

47. Comment: L429-433: You state that the method you developed also applies for the rain-fed agriculture. This is correct, because then you just exclude the irrigation parameter from the SWAT model. Nevertheless, this is a commonly used method and I do not see any novelty of your method here. For this reason, I recommend to delete this section. Response: Thank you for your comments. We have deleted these sentences in the revised manuscript as suggested.

48. Comment: L456-457: You already stated in the L45, that blue water has the largest part of the total WF. Please delete this sentence. Response: Thank you for your comments. We have deleted these sentences in the revised manuscript as suggested.

Thank you for your helpful suggestion regarding our manuscript. We have revised the manuscript according to your comments carefully. We hope these modifications, based on your suggestions, will raise the quality of our manuscript to meet the publication standards of Hydrology and Earth System Sciences. We appreciate the editors and reviewers' work. Once again, thank you very much for your comments and suggestions.

Please also note the supplement to this comment:
https://www.hydrol-earth-syst-sci-discuss.net/hess-2018-125/hess-2018-125-AC1-

supplement.pdf

[Figure]

Map legend:

- Irrigation diversion intake
- Weather station
- Irrigation canal
- Yellow River
- Study area
- Hetao irrigation district
- Wuliangsuhai Lake

County
- Dengkou
- Hangjinhouqi
- Linhe
- Wuyuan
- Wulateqianqi
- Wulatezhongqi
- Wulatehouqi

Map labels: Wulatehouqi, Wulatezhongqi, Wuyuan, Linhe, Wulateqianqi, Hangjinhouqi, Dengkou

0 10 20 30 40 Kilometers

**Fig. 1.** Fig. 1. Location of the Hetao Irrigation District (HID) in China

Main canal irrigation area
$(S_j , k_j , Q_j)$

Point $(Q_{ji})$

Zone $(S_{jn}, W_{jn})$

$n$

$i$

$2$

$1$

Main canal
$(k_{mc})$

$1$  $2$  $3$  $\cdots$  $j$  $\cdots$  $m\text{-}1$  $m$

River  General main canal $(k_{gc})$

**Fig. 2.** Fig. 4. Model for calculation of water loss in canal system

---

## Author Comment (AC2) · 4 Jul 2018

<Manuscript number: HESS-2018-125> Dear Editors and Reviewers: Thank you for your letter and for the reviewers' comments concerning our manuscript "An improved method for calculating regional crop water footprint based on hydrological process analysis". We appreciate your comments and constructive suggestions very much, and they were valuable for improving the quality of our manuscript. We have revised the manuscript in detail according to the editor and reviewers' comments. We hope that these modifications, based on your suggestions and the reviewers' comments, will raise the quality of our manuscript to meet the publication standards of Hydrology and

Earth System Sciences. The revised portions are marked in red in the paper. The main corrections in the paper and the responses to the reviewer's comments are as follows:

Anonymous Referee #2

SUMMARY AND GENERAL COMMENTS The paper presents a semi-distributed approach to model effective water resource requirements in crop production in terms of the volume of water used per unit crop production. The approach differentiates between green and blue water sources and puts emphasis on conveyance losses of irrigation water. Modelling of the water cycle is based on SWAT, while conveyance losses between the water inlet of the irrigation scheme and the field are modelled depending on the location according to a new approach that, apparently, has not been published before. The novel contribution to the field of science by this study is limited to the location-dependent modelling of conveyance losses, which can potentially have significant effect on crop water footprint calculations. Unfortunately, the derivation of the approach is neither explained in much detail nor is its validity tested against measured data. Overall, the presentation of the theoretical background, methods and results is rather poor and, at least partly, hard to understand. The language is unprecise and redundant in major parts of the paper. It leaves room for interpretation (eg lines 64-66) and numerous sentences/paragraphs are unintelligible (e.g. lines 86-87, 90-93, 104-105, 207-209). I am not a native English speaker but I feel the text needs revision with regards to pure language issues (grammar, mode of expression). The paper does not provide a critical discussion of the approach and the results. In particular, uncertainties of inputs and results are hardly addressed. Major parts of the discussion section basically repeat the contents of the introduction. The conclusions section is basically a summary of the results and the few conclusions made are trivial. The title does not match the content of the manuscript (see comment on the term "water footprint" below). Response: Thank you for your comments. The main purpose of this study is find a better method to quantify crop production water footprint more comprehensively, because the current method of crop production water footprint does not fully contain

all the water consumed in the crop production process, such as water loss from the channel when the water is transported to the field. In this study, we considered the water consumption associated with crop production in general, including canal water loss, which has not been studied by previous studies. At the same time, the spatial resolution of the results of the crop production water footprint in this study is higher, and the water footprint changes within the region can be found. These contribute to a truly quantified crop production water footprint, more accurately assessing the crop production water footprint in a given area, and thus more precise determination of water footprint hotspot areas. This will provide the basis for water resources management in the region. According to your comments, we have modified the thesis in the revised manuscript as suggested. At the same time, we have improved the language of the thesis (The paper was edited by Elsevier Language Editing Services).

DETAILED COMMENTS ON SUBSTANTIAL SHORTCOMINGS OF THE MANUSCRIPT 1. Comment: The authors refer to the water resource requirements of crop production as "water footprint", which is inappropriate two reasons. Firstly, indirect water uses, an important aspect of a footprint indicator, are not considered in the study. Secondly, the paper lacks a clear definition of the system (consumer or producer) that causes the footprint. Response: Thank you for your comments. 1. Crop production consumes plenty of water resource. Fertilizers, pesticides and machinery also contain indirect water footprints. Due to the lack of above data, to quantify water footprint of crop production in the world mainly focus on the evaluating water use during crop production, which is the direct water footprint. The reference are as follows: Bocchiola, D., Nana, E., & Soncini, A. (2013). Impact of climate change scenarios on crop yield and water footprint of maize in the Po valley of Italy. Agricultural Water Management, 116(2), 50-61. Cao, X., Wu, P., Wang, Y., & Zhao, X. (2014). Water Footprint of Grain Product in Irrigated Farmland of China. Water Resources Management, 28(8), 2213-2227. Hoekstra, A.Y., & Mekonnen, M.M. (2012). The water footprint of humanity. PNAS, 109(9), 3232-3237. Mekonnen, M.M., & Hoekstra, A. Y. (2011). The green, blue and grey water footprint of crops and

Interactive
comment

derived crop products. Hydrology & Earth System Sciences, 15(5), 1577-1600. Zhuo, L., Mekonnen, M. M., & Hoekstra, A. Y. (2016). Benchmark levels for the consumptive water footprint of crop production for different environmental conditions: a case study for winter wheat in China. Hydrology & Earth System Sciences Discussions, 20(11), 4547-4559. 2. As for your second suggestion, we have explained in this manuscript that the water footprint in this study is generated during crop production.

2. Comment: The paper presents water resource requirements for the production of three different crops (m3 water use/t of crop production, referred to as "water footprint") in subbasins of the Hetao Irrigation District (HID). Obviously, the "water footprint" is defined for a producer. It is not stated whether the footprint figures are calculated for (a) a single producer, i.e., the aggregate of "farms" growing a single crop type in the HID, or (b) many different producers, i.e., the aggregates of farms growing that crop within individual subbasins. However, this is important in order to understand the results correctly. In case (a) the volume of water used to produce $x_i$ tonnes of crop in subbasin $i$ needs to be related to the total crop production in HID ($X$). If $r_i$ is the water resource requirement in sub-basin $i$, the water footprint of the HID-wide crop production in subbasin $i$ calculates as $F_i=x_i/X*r_i$. In contrast, the water footprint of subbasin-wide crop production (case (b)) in subbasin $i$ is given as $F'_i=r_i$. Note that in case (b), the "water footprint" indicator is no longer geographically explicit, another important aspect of the water footprint, as the subbasins are the smallest geographical units presented. The range of results shown in the maps implies that the water footprint is defined according to case (b). However, water resource requirements for crop production are intrinsic properties of the irrigation system in each subbasin and are independent of the actual allocation of crop production. Hence, the study is not a footprint analysis but, simply, an analysis of resource requirements (comparable to a potential analysis). However, the representativeness of the results is questionable due to methodological limitations. Subbasins are sub-divided into hydrological response units (HRU) based on land use (supposedly land use=crop type) and soil type. Although it is not stated explicitly, one must assume that the results on HRU-level, based on the actual pattern of crop allocation and irrigation timing/quotas, are aggregated to subbasin-level (aggregation method not specified). This way, the results are only representative for potentially small parts of a subbasin, i.e., one or more HRUs within a subbasin under the given crop, as the conditions (soil type, canal losses, etc.) may be different in the remaining parts of the subbasin. The reader cannot judge the related uncertainties as the actual patterns of crop allocation and soil types are not shown. Response: Thanks you for your comments. In this study, water footprint during crop production was calculated by SWAT model. SWAT model divides the region into subbasins according to DEM and water system. Then subbasins are divided into HRU according to land use type, soil type and slope. Among them, subbasin is the smallest geographic unit. Therefore, the steps to calculate the water footprint during crop production are as follows: At first, water consumption for a certain crop in a certain subbasin was calculated, that is, the total water consumption of each HRU. Then crop production in each HRU was calculated (HRU area multiplies crop yield per unit area). Then total water consumption was divided by total crop production in this subbasin to obtain the water footprint during crop production in this subbasin. This study mainly focused on analysis on water use in irrigated area, and irrigation water loss during convey was taken into consideration. The results of this research have improved spatial resolution with more detailed reflection of water footprint changes inside the region compared to former researches, which is of vital significance to local water resource management. Due to the limited resolution of land use data, the specific distribution of each crop in study area couldn't be distinguished. Additionally, in the Hetao irrigation district, farmers generally plant three crops (wheat, corn, sunflower) to diversify their business risk. Because of the large population, the total farmland of each farmer is small. Therefore, three crops are evenly distributed as a certain proportion in the entire HID. As a result, in the SWAT model's HRU partition setting, we further divide agricultural land into small parts by Land Use Refinement tab, which depends on the proportion of three crops, so that the SWAT model can distribute three crops evenly across the irrigated area. It is an important reference to divide the HRU. The SWAT model distributed all three crops proportionally to the irrigated area,

and land use types might influence the calculation of water footprint results. In addition, there were more than ten types of soil in this study area, but the influence of soil types on water footprint is not clear. Therefore, the uncertainties of land use types and soil types on water footprint have not been taken into account in this study, but you have provided us with a new ideas for future study, thank you very much.

3. Comment: The description of the methods to calculate the "water footprint" is difficult to understand. As the system boundaries are not defined precisely, the reader is forced to examine several possible system boundaries in order to judge whether the equations 6-9 are likely to be correct. For instance, it depends on the sys-tem boundary whether field discharge (Qd) is actually consumption, i.e. it is a flow out of the system (to another basin or the sea), or returns to system itself. As the authors stress that the approach is regional-scale, a certain share in field discharge is likely a return flow, which would invalidate equation 7, which defines field discharge as water consumption. Equations 6-9 use a set of variables that are calculated for two different scenarios (s1=with irrigation, s2=without irrigation) but the notation is ambiguous as the scenario is not clearly indicated in the equations except for for ET (index s1 or s2). It might be considered obvious that canal losses (Qc) and ET of field irrigation (Qf) is only defined for the scenario with irrigation (s1). (Note, those variables can also be defined for s2, though with a value of zero.) However, capillary rise of groundwater (Qg) and field discharge (Qf) definitely can have non-zero values for s2. Hence, it must be indicated from which scenario the values are taken. Qg must not be added in eq 7. Although Qg is per definition blue water, it simple changes soil moisture. The share of Qg that is consumed is already included in Qf+Qd. Response: Thanks you for your comments. In this study, crop production was obtained by local statistic data, and the green and blue water were obtained by SWAT model. Due to the little precipitation which was difficult to meet the growing needs of crops, green water consumption was equal to the effective precipitation. In this study, we have set scenario 2 without irrigation to calculate the effective precipitation (formula 6). The blue water consumption includes water loss in canal system, the consumption of irrigation water in the field, and the drainage in

the field. Water loss in canal system during convey was obtained by irrigation water consumption in the field which was obtained by SWAT model and effective utilization coefficient of canal water. The consumption of irrigation water in the field was the ET of irrigation water, obtained by total ET minus green water consumption. The drainage in the field was the extra water discharged from the field, obtained by the outputs of SWAT model (formula 10). According to your comments, we have modified the description on calculation process. In this study area (HID), drainage in the field eventually flowed out of the irrigated area and could no longer be used in the irrigated area. Therefore, in this study drainage in the field was part of the blue water consumption. Of course, for other areas, if drainage in the field didn't flow out of the irrigated area and could be reused, then the field drainage was not blue water consumption, which needed to be clarified before calculation. The situation of each indicator in the formula is indeed not clearly pointed out. According to your suggestion, we added the situation of each indicator in the formula. We fully agree with your opinion. Qf+Qd has included Qg. During crop growth, the sources of ET are precipitation, soil water and groundwater. If groundwater rises into the soil, it is consumed by crop evaporation or drainage. We have revised the formula according to your suggestion. Due to the sufficient amount of irrigation in the study area, there was little use of groundwater, which have little impact on this research. The modified parts are as follows: Page 11, 12, line 198-226. Water consumption in the fields consists of 4 parts including the actual ET of precipitation, irrigation water, groundwater utilized by crops, and field drainage. This study set up two scenarios and calculated the above water consumption by changing the sources of water in the SWAT model. In scenario 1 (S1), crop water consumption was derived from precipitation and irrigation water (irrigation systems and irrigation quotas are based on local irrigation methods), i.e., the actual situation of crop water use. In scenario 2 (S2), crop water consumption was only derived from precipitation without irrigation. The S2 was used to calculate the consumption of green water. In this study area (HID), because of less rainfall, the effective precipitation formed by precipitation is all used for crop growth. Therefore, the consumption of green water for crops is equal

to the effective precipitation, which means that green water is reflected by calculating the effective precipitation stored in soil by SWAT model. The calculation formula is as follows. (5) (6) (7) (8) (9) (10) where WF is the water footprint of crop production (m3/t), WFg is the green footprint (m3/t), WFb is the blue water footprint (m3/t), Wg is the green water consumption during the crop growth period (m3), Wb is the blue water consumption during the crop growth period (m3), Y is the crop yield (t), PRECIPs2 is the precipitation during the crop growth period in Scenario 2 (m3), SUPQs2 is the surface runoff during the crop growth period in Scenario 2 (m3), LATQs2 is the soil lateral flow during the crop growth period in Scenario 2 (m3), Qc is the amount of water loss in the canal system (m3), Qf is the actual ET of field irrigation water (m3), Qd is the field discharge (m3), It,s1 is the total amount of irrigation water diversion in Scenario 1 (m3), and If,s1 is the actual amount of water irrigated in the field in Scenario 1 (m3). ks1 is the effective utilization coefficient of canal water in Scenario 1(Obtained from the local Water resources management department), ETs1 is the crop actual ET during the crop growth period in Scenario 1 (m3), WYLDs1 is the total amount of water leaving the HRU in Scenario 1 (m3). The data of parameters PRECIPs2, SUPQs2, LATQs2, It,s1, ETs1, WYLDs1 were obtained from the SWAT model.

4. Comment: As I understand, canal losses in eq 7-8 are informed by the modelling approach represented by eq 10-15 but it remains unclear which of the variables mentioned in eq 10-15 are actually used and how. The notation of eq. 10-15 is confusing as I suspect most readers are familiar with a notation where n is the total number of elements and i is a running index. Here, it is used the other way around, which is not wrong but makes it more difficult to understand. Response: Thanks you for your comments. In this study area (Hetao irrigation district), irrigation canal system is complicated, which is totally seven levels from big to small, and water exists in the general main canal and the main canal during crop growth period (for timely irrigation for crop), while in other channels water only exists in irrigation periods. Thus, according to the characteristics of canal system in the study area, we have divided canal system into two parts to calculate the total water loss, in which Part A is water loss in the general

main canal and the main canal, and Part B is water loss in other canals. According to the simplified model in the paper and the methods of interpolation in ArcGIS software, the two parts of water loss were distributed throughout the irrigation area respectively, then the two interpolation results were added to obtain water loss distribution in this irrigation canal system. In formula 10-15, Wa and Wb are the total amount of canal system water loss of Part A and Part B respectively. The variables kgc and kmc are the canal system water utilization coefficient of the general main canal and the main canal respectively, which are used to calculate the canal system water loss of part A. Qji and Qj are water loss of the canal system per unit area in Part A and Part B used for interpolation in ArcGIS software. Sj is the area that a certain canal can irrigate. n in formula Qn is the total number of elements. The symbol of this formula is not written correctly. Thank you for your advice and we have modified this formula. The modified parts are as follows: Page 13-16, line 228-280. Water transfer loss is a kind of water loss in the process of channel water delivery, and it is an important part of blue water consumption in crop production. For a piece of cultivated land, the water loss during the process of the crop production includes the loss of water from the water source to the field flowing through the canal system. In the Hetao Irrigation District, irrigation canal is composed of seven grades (general main canal, main canal, sub-main canal, branch canals, lateral canals, field canals, and sub-lateral canals). Because of the complex distribution of canal system and the lack of hydrological data in irrigation districts (the lack of effective utilization coefficient of canal water below the main canal). Therefore, in calculating the water loss of canal system during crop production process, we generalized Hetao Irrigation District into a model similar to the histogram (Fig. 4). We divide the total water loss of canal system into two parts. Part A is the loss of the main canal and canal, and Part B is the loss of the remaining canal system (the water loss of the sub-main canal and its sub-channels at all levels). The calculation of water loss in part A is as follows: first, the water loss of each section is calculated by dividing the main canal into equal distances (10 km). Then the water transfer loss of each section of the canal is allocated to each field downstream [Equation 10], thereby obtaining the water

[Figure]

transfer loss in the crop production process on the field block. Therefore, the actual water loss caused by irrigation in a field is the sum of the water loss of the transfer canal and the canal in the upstream. We assign the actual water loss of the field by irrigation (Qji, formula 11) to the midpoint of the each section, and use Kriging interpolation in ArcGIS to obtain the water loss distribution map of the figure a (Part A). Due to the lack of the effective utilization coefficient of canal water and the distribution map of the canals at all levels and below, the calculation process of the water loss in Part B is as follows: the remaining canal loss in each irrigation canal is divided by the main canal irrigation and the unit area loss of the canal control area is obtained. Then, the amount of water loss per unit area within the control range of each main canal in the irrigation area (Qj, formula 15) is obtained, and the data is brought into ArcGIS for the water loss distribution map of figure b (Part B). Finally, the figure a and the figure b are superimposed and calculated in the ArcGIS using the map algebra module of the spatial analysis tool to obtain the water loss distribution map of the canal system in HID. The formulas are as follows: (11) (12) (13) (14) (15) (16) where Qji is the actual amount of water loss per unit area of the i section of the jth main canal (m3/ha), Wjn is the water loss per unit area of the section of the jth main canal in part A (m3/ha), j is the number of the main canal, i is the number of the equidistance sections in the jth main canal, n is the total number of the sections in the jth main canal, m is the total number of the main canals, WA is the amount of water loss in part A (m3), kj is the coefficient of the water distribution from the general main canal to the jth main canal, Sjn is the area of each sections in the jth main canal (ha), It,s1 is the amount of total irrigation water diversion in Scenario 1(m3), kgc is the water conveyance efficiency of the general main canal, kmc is the water conveyance efficiency of the main canal, Sj is the area controlled by the jth main canal (ha), Qj is the water loss per unit area of the jth main canal (m3/ha), WB is the amount of water loss in part B (m3), and Qc is the amount of water loss in the canal system (m3).

Fig. 4. Model for calculation of water loss in canal system Note: Sjn is the area of each sections in the jth main canal, Wjn is the water loss per unit area of the section

of the jth main canal in Part A, Qji is the actual amount of water loss per unit area of the i section of the jth main canal, Sj is the area controlled by the jth main canal, kj is the coefficient of the water distribution from the general main canal to the jth main canal, Qj is the water loss per unit area of the jth main canal in Part B, kgc is the water conveyance efficiency of the general main canal, kmc is the water conveyance efficiency of the main canal, j is the number of the main canal, i is the number of the equidistance sections in the jth main canal.

5. Comment: The section on calibration and validation of the model is wordy and interrupts the description of the modelling approach. For instance, the R2 metric is widely used and there is no need to show the formula. If equations 2-4 are considered necessary, the notation should be corrected as the index i is missing in numerous terms. Response: Thank you for your comments. According to your suggestion, we have deleted the corresponding formula. And for more perfect presentation, we have put calibration and validation of SWAT model section to supplement information. The modified parts are as follows: Page 15, 16, line 178-186. 2.4 Calibration and validation The Sequential Uncertainty Fitting (SUFI-2) algorithm in SWAT-CUP was applied for calibration and validation (Abbaspour et al., 2007; Abbaspour, 2012) by comparing the simulated stream discharge from the model with the measured discharge data. The global sensitivity analysis integrated within SUFI-2 was used to evaluate the hydrologic parameters for the discharge simulation and then the optimal simulation is established by adjusting the sensitivity parameters and through multiple iterations. The calibration period was from 2006-2009, and the validation period was from 2010-2012. The result of the SWAT calibration and validation process is satisfactory, the detailed process are available in support information.

CONCLUSIONS Given the shortcomings addressed above, the quality of the manuscript is, in my opinion, not acceptable for publication, although the underlying material fits the scope of the journal and might be worth publishing. Due to missing definitions and precise description of the methods, I can hardly judge the validity of the

work. I think the necessary revisions are too extensive to be done within a peer-review process. Apart from this, addressing all the issues where I see the need for revision in this reviewer comment would be an unreasonable effort. Therefore, my recommendation is to reject the paper. Response: Thanks for your careful review of this paper and constructive suggestions. According to the comments, major revisions have been done which are as follows: 1. In the revised manuscript, detailed description has been added about methods and parameter in this paper. 2. The innovations of the research have been restated

Thank you for your helpful suggestion regarding our manuscript. We have revised the manuscript according to your comments carefully. We hope these modifications, based on your suggestions, will raise the quality of our manuscript to meet the publication standards of Hydrology and Earth System Sciences. We appreciate the editors and reviewers' work. Once again, thank you very much for your comments and suggestions.

Please also note the supplement to this comment:
https://www.hydrol-earth-syst-sci-discuss.net/hess-2018-125/hess-2018-125-AC2-supplement.pdf
* * *
Main canal irrigation area
$(S_j , k_j , Q_j)$

Point $(Q_{ji})$

Zone $(S_{jn}, W_{jn})$

$n$

$i$

Main canal
$(k_{mc})$

1    2    3    ...    $j$    ...    $m$-1    $m$

River   General main canal $(k_{gc})$

**Fig. 1.** Fig. 4. Model for calculation of water loss in canal system

---

## Author Response (AR2)

**\<Manuscript number: HESS-2018-125\>**

**Dear Editors:**

Thank you for your comments concerning our manuscript "An improved method for calculating regional crop water footprint based on hydrological process analysis". We appreciate your comments and constructive suggestions very much, and they were valuable for improving the quality of our manuscript.

We have revised the manuscript in detail according to your comments. The revised portions are marked in red in the paper.

The main corrections in the paper and the responses to the editor comments are as follows:

**Comment 1:**

Section 4.4 "The influence on efficiency of irrigation system" as stated in the response letter to Referee#1 has become the first paragraph in section 4.3 "Strategies for adjusting the crop production water footprint" in the main manuscript. I'd like to ask the authors to rethink this shift and suggest to keep section 4.4 (on efficiencies) or to move this paragraph to the end of section 4.3.

**Response:**

Thank you for your comment. We have moved "The influence on efficiency of irrigation system" paragraph to the end of section 4.3 in the revised manuscript as your suggestion. The efficiency of irrigation system is one of the factors that affect the crop production water footprint.

**The modified parts are as follows:** (Page 23-24, line 414-438)

**4.3 Strategies for adjusting the crop production water footprint**

The water footprint of crop production is affected by crop species. Different crops have different water use characteristics and different growth periods. Therefore, adjusting the crop planting structure can change the water supply in the region (Fasakhodi et al., 2010), which in turn affects the water footprint of crop production. At the same time, changing the crop pattern, planting crops which growth periods are consistent with the precipitation period can increase the utilization of green water, reduce the consumption of blue water, and reduce the pressure on local water resources (Liu et al., 2018). This study found that in the HID, the growth period of sunflower is basically the same as the precipitation period. Consequently, expanding the planting area of sunflower can make better use of local precipitation resources and reduce the use of blue water.

Crop yield is an important factor affecting the water footprint of crop production. Selecting crop varieties with high yields and improving agricultural management measures play an important role in increasing crop yields. Sun et al. (2013b) found that improving agricultural management measures is an important factor to increase crop yield and reduce water footprint of crop production. Liu et al. (2014, 2015) discussed the water use situation and virtual water flow in Hetao Irrigation District and found that crop yield had an important impact on the water footprint of crop production, and with the increasing of crop yield per unit area, the water footprint of crop production had declined.

The efficiency of irrigation system is affected by the way of water transportation, the condition of canal system, the irrigation technology and so on. Therefore, the water use efficiency of the regional irrigation system can be improved by changing the water delivery method (from the channel to the pipeline) and the irrigation method (such as dropper, sprinkler and other advanced irrigation technologies). For the study area, the results show that more than half of the water resources were lost during the process of canal water transport and irrigation. Therefore, adopting anti-seepage measures to reduce the leakage of canal system, and adopting advanced irrigation technology to reduce the amount of irrigation water will help to reduce the water footprint of crop production in this region.

**Comment 2:**

Please use the same units when comparing your results with results from other studies.

**Response:**

Thank you for your comment. We have modified the units ("$m^3$ $kg^{-1}$" change to "$m^3$ $t^{-1}$") in the revised manuscript as your suggestion.

**The modified parts are as follows:**

Sun et al. (2013b) calculated the average water footprint of HID by using regional scale method and water balance principle and the result was 3910 $m^3$ $t^{-1}$. (Page 22-23, line 400-402)

Qin et al. (2016) calculated the water footprint of sunflower in Jilin province by using field scale method and found that the water footprint of sunflower in this area from 2006 to 2008 were 1280 $m^3$ $t^{-1}$, 1684 $m^3$ $t^{-1}$ and 1726 $m^3$ $t^{-1}$, respectively, which was smaller than this study. (Page 23, line 404-406)

Thank you for your helpful suggestion regarding our manuscript. We have revised the manuscript

according to your comments carefully.

We appreciate the editors and reviewers' work.